



# Characterizing groundwater heat-transport in a complex lowland aquifer using paleo-temperature reconstruction, satellite data, temperature-depth profiles, and numerical models.

Alberto Casillas-Trasvina[1,2*], Bart Rogiers[1], Koen Beerten[1], Laurent Wouters[3], Kristine Walraevens[2]

[1]Institute for Environment, Health and Safety. Belgian Nuclear Research Centre, SCK CEN, Boeretang 200, 2400 Mol, Belgium.
[2]Laboratory for Applied Geology and Hydrogeology, Department of Geology, Ghent University,  Krijgslaan 281-S8, B-9000 Ghent, Belgium.
[3]ONDRAF/NIRAS, Belgian Agency for Radioactive Waste and Enriched Fissile Materials, Kunstlaan 14, B-1210 Brussels, Belgium.

*Correspondence to*: Alberto Casillas-Trasvina (jesuscasillasmx@outlook.com)

**Abstract.** Heat is a naturally occurring widespread groundwater tracer that can be used to identify flow patterns in groundwater systems. Temperature measurements, being relatively inexpensive and effortless to gather, represent a valuable source of information which can be exploited to reduce uncertainties on groundwater flow, and e.g. support performance assessment studies on waste disposal sites. In a lowland setting, however, hydraulic gradients are typically small, and whether temperature measurements can be used to inform us about catchment-scale groundwater flow remains an open question. For the Neogene aquifer in Flanders, groundwater flow and solute transport models have been developed in the framework of safety and feasibility studies for the underlying Boom Clay Formation as potential host rock for geological disposal of radioactive waste. However, the simulated fluxes by these models are still subject to large uncertainties, as they are typically constrained by hydraulic heads only. In the current study we use a state-of-the-art 3D steady-state groundwater flow model, calibrated against hydraulic head measurements, to build a 3D transient heat-transport model, for assessing the use of heat as an additional state variable, in a lowland setting, at the catchment scale. We therefore use temperature-depth (TD) profiles as additional state variable observations for inverse conditioning. Furthermore, a Holocene paleo-temperature time curve was constructed based on paleo-temperature reconstructions in Europe from several sources in combination with land-surface temperature (LST) imagery remote sensing monthly data from 2001 to 2019 (retrieved from NASA's MODIS). The aim of the research is to understand the mechanisms of heat transport and to characterize the temperature distribution and dynamics in the Neogene aquifer. The simulation results clearly underline advection/convection and conduction as the major heat transport mechanisms, with a reduced role of advection/convection in zones where flux magnitudes are low, which suggests temperature is a useful indicator also in a lowland setting. Furthermore, performed scenarios highlight the important roles of i) surface hydrological features and withdrawals driving local groundwater flow systems, and ii) the inclusion of subsurface features like faults in the conceptualization and development of hydrogeological investigations. These findings serve as a proxy of the influence of



advective transport and barrier/conduit role of faults, particularly the Rauw Fault in this case, and suggest that solutes released from the Boom Clay might be affected in similar ways.

## 1 Introduction

Heat is a naturally occurring widespread groundwater tracer that can be used to identify flow patterns in groundwater systems (Anderson, 2005; Bense et al., 2017; Saar, 2011). Yet, it is often not evenly distributed in basins as thermal heterogeneities are observed in the increase of temperature with depth (Dentzer et al., 2017). Recent work has broadened the use of heat in a quantitative way by incorporating it in formal solutions of the inverse problem to estimate hydraulic properties and groundwater flux (Hecht-Méndez et al., 2010; Jiang and Woodbury, 2006; Liu et al., 2019; Munz et al., 2017; Rau et al., 2010;

Rodríguez-Escales et al., 2020). This is commonly done with numerical codes that at least enable one-way coupling of the different processes, i.e. groundwater flow and heat transport.

Groundwater flow induces heat advection, being a significant component of the total heat flux especially in sedimentary basins, and thereby influencing the subsurface temperature distributions for relatively deep groundwater flow (Anderson, 2005; Pollack et al., 1993; Saar, 2011). Due to this thermal signature transmitted by the movement of groundwater, an analysis of

subsurface temperature distributions can yield quantitative insight into the groundwater flow systems behavior (Saar, 2011). Additionally, when used as a tracer, groundwater temperatures are more sensitive to, for instance, the connectivity patterns and fault zones within an aquifer compared to hydraulic data alone, providing supplementary information on aquifer structure (Bense et al., 2017; Kurtz et al., 2014; Read et al., 2013). Temperature measurements, being relatively inexpensive and effortless to gather, hence, represent a valuable source of information which can be exploited to constrain groundwater flow

(Anderson, 2005; Irvine et al., 2017; Kurylyk et al., 2019; Saar, 2011). However, as stated by Schilling et al. (2019), it is currently an underrepresented state variable in groundwater studies, with respect to hydraulic head. Schilling et al. (2019) present a robust review on the use of what they call 'unconventional' state variables, including temperature observations. They mention that the inclusion of temperature observations in combination with 'conventional' observations (i.e. hydraulic head) is beneficial for heat-transport simulations, given that heat-transport is not appropriately simulated on its own (Schilling et al.,

2019). However, in many cases, and as mentioned by Bravo et al. (2002), Kurtz et al. (2014), Irvine et al. (2015) and Delsman et al. (2016), when there are multiple unknowns on flux exchanges and thermal and hydraulic properties, it is likely to reproduce temperature correctly in spite of a potential incorrect representation of fluxes (Schilling et al., 2019). Nonetheless, most of these studies (i.e. where temperature observations have been implemented to evaluate aquifer hydraulic characteristics) have been performed in shallow environments mainly for surface water/groundwater interactions/exchanges at depths of

around 20 m (Bartsch et al., 2014; Bravo et al., 2002; Delsman et al., 2018; Engelhardt et al., 2013; Kurtz et al., 2014; Liu et al., 2019; Ma et al., 2012; Munz et al., 2017; Rau et al., 2010; Read et al., 2013; Shanafield and Cook, 2014; des Tombe et al., 2018) with the exception of the works of Masbruch et al. (2014) and Irvine et al. (2015) which present works for deeper environments (i.e. few hundreds of meters). Cases where temperature observations have been taken at relatively large depths





(i.e. a few hundreds of meters) and used as observations for heat-transport simulations are scarce (Masbruch et al., 2014).
Several studies where temperature profiles are measured are typically used for qualitative interpretations of the effects of anthropogenic stressors (Benz et al., 2018; Dong et al., 2018) and for evaluation of deep (i.e. hundreds to thousands of meters) geothermal activities where substantial temperature variations occur (Dentzer et al., 2017; Majorowicz and Grasby, 2020; Marty et al., 2020; Sippel et al., 2013; Smith and Elmore, 2019). Little research has however been devoted to exploring the use of temperature in aquifers with depths in the range of a few tens of meters, up to a few hundred meters, in which the
temperature range is rather limited (up to a maximum of ±10 degrees). Moreover, in a lowland setting, hydraulic gradients are typically smaller, and it is unclear whether advection of heat at the catchment scale can be sufficiently large to enable the use of temperature.

For the Neogene aquifer in Flanders, groundwater flow and transport models have been developed in the framework of safety and feasibility studies for the underlying Boom Clay Formation as potential host formation for geological disposal of
radioactive waste (Gedeon, 2008; ONDRAF/NIRAS, 2010, 2013; Rogiers et al., 2015; Vandersteen et al., 2013). However, the simulated fluxes by these models are still subject to large uncertainties, as they are typically constrained by hydraulic heads only. While the evaluation of candidate host formations continues, this study investigates how heat transport is affected by groundwater flow in the sedimentary Neogene aquifer, across the Nete catchment, in Belgium. Additionally, emphasis is put on the disturbances in heat transport by the presence of faults, highlighting the Rauw Fault – a 55 km long normal fault. To
this end, we use the state of the art 3D groundwater flow model presented by Casillas-Trasvina et al., (accepted for publication 2021), gathered temperature-depth (TD) profiles spread over the catchment, land-surface temperature (LST) satellite images data (NASA's MODIS), and a paleo-temperature time series to set a transient temperature boundary condition. This case study will help assessing the usefulness of temperature data in a catchment-scale lowland setting to characterize the magnitude and patterns of groundwater flow. This approach seems especially suitable within the framework of the disposal of radioactive
waste, as the idea is to learn as much as possible from measurements from low to non-invasive techniques This work serves as a case study attempting to integrate the information provided by temperature data as additional 'unconventional' state variable at the catchment scale in a quantitative way, with the objective of further constraining the numerical models that serve as summaries of our system understanding.

## 2 Study area and hydrogeological setting

The Neogene aquifer is located in the Campine area, in the northeast of Flanders and is considered to be the most important groundwater reservoir in the region (Coetsiers and Walraevens, 2006). A full description of the study area has been presented elsewhere (Casillas-Trasvina et al., (accepted for publication, 2021)). For clarity, only a brief summary is given here. The area of the Neogene aquifer within the Nete catchment is studied in this work, as shown in Figure 1. The area is characterized by a low relief with altitudes ranging from ~5 to 70 meters (above TAW, Tweede Algemene Waterpassing) along a west-east
gradient. As a result, the hydrography is characterized by an east-west drainage system that belongs mainly to the Scheldt



River basin (Van Keer et al., 1999; Rogiers et al., 2014). The aquifer is mainly composed of Neogene marine sand deposits that induce some variations in hydrochemical composition, although the groundwater is weakly mineralized (Coetsiers and Walraevens, 2008). The Oligocene (Rupelian) and Mio-Pliocene geology of the study area is presented in Figure 2. The lithology mainly consists of fine to medium grained sands, while clay content is found to vary in certain units (e.g. Kasterlee, Lillo, and Diest formations) and basal gravels are sometimes present between the units (Laga et al., 2001). The sediments dip towards the north-northeast with a gentle slope of about 1-2% with some disturbances towards the west by different normal faults. In the eastern/north-eastern part of the study area faults occur that were formed as a consequence of the development of the Roer Valley Graben (RVG), the northwestern-most part of the Lower Rhine Graben (LRG) (Verbeeck et al. 2017; Deckers et al. 2018). The most important of these faults outside the proper RVG, and in terms of Cenozoic offset, is the Rauw Fault which is proven to have been active during the Pleistocene (Verbeeck et al., 2017). The Rauw Fault has a displacement of more than seven meters in the Quaternary, which increases with depth. It does, however, not have a clear surface expression. The Rauw Fault consists of two separate branches (i.e. Rauw Fault, and Rauw East), around 700 m apart, for which the movement of the Rauw East fault has stopped earlier and Rauw Fault has taken over the activity (Verbeeck et al., 2017). The observed hydraulic gradient across the Rauw Fault appears significant, with head differences of 1.5-2.0 meters over a horizontal distance of around 60 meters as observed in the ON-MOL-2 site in the municipality of Mol, in the northeast of Belgium (Coordinates for the ON-MOL-2A: E 5°11'54.12"; N 51°14'54.04"). For a more detailed description please refer to Verbeeck et al. (2017).

Quaternary deposits of varying texture overlie the Neogene units and constitute the upper few meters of the aquifer system. The hydrostratigraphical units occurring below the Quaternary are composed of Pleistocene and Pliocene sediments. These consists of the Malle, Merksplas, Mol and Lillo Sands, sitting on the Kasterlee Formation, a mixed clayey-sandy formation deposited in shallow-marine to estuarine conditions. It is followed by the Diest Sands, overlaying the Lower Miocene Berchem Sands and Late Oligocene Voort Sands. The Boom Clay, an Oligocene marine sediment, forms the lower boundary of the system. For a more detailed description of the hydrostratigraphy of the area and the Boom Clay, please refer to Laga et al. (2001), Coetsiers and Walraevens (2008), Yu et al. (2013) and Vandenberghe et al. (2014). Notwithstanding their lithological differences, Patyn et al. (1989) concluded from hydrogeological observations that these sediments behave as a single aquifer. Similarly to Casillas-Trasvina et al., (accepted for publication, 2021), in this work, the combined Quaternary deposits, Pleistocene, Upper and Lower Pliocene aquifers together with the Lower Miocene and Oligocene Aquifer System are referred to as the 'Neogene Aquifer'.

Various subsurface and surface activities are taking place or being planned above and below the Neogene aquifer, including but not limited to surface and geological disposal of low-level and short-lived and long-lived nuclear waste. The Nete catchment and the Neogene aquifer have been subject to studies in the framework of geological disposal of nuclear waste (Beerten et al., 2010; Gedeon, 2008; Mallants, 2010; Mazurek et al., 2009; ONDRAF/NIRAS, 2010, 2013).



## 3 Methodology

### 3.1 Data collection

#### 3.1.1 Temperature-depth (TD) profiles

Temperature-depth (TD) profiles have been collected in the framework of various projects and campaigns. Historical data was compiled by Rogiers et al. (2015) from previous campaigns, presented in Table 1. The quality of the TD profiles taken from 1988 to 1997 is unknown since sensor resolution and accuracy are not known together with the performed logging speed (up to four logs per day), potentially inducing considerable bias in the obtained temperatures. The location of these TD profiles is

presented in Figure 2. The TD profiles collected during the site characterization for the cAt project (ONDRAF/NIRAS, 2010) were not considered during the assessment by Rogiers (2014) since they were believed to be influenced by the drilling activities, and thermal equilibrium wasn't reached yet.

Four TD profiles reported by Rogiers (2014) were the only data for which type and characteristics of sensor and  logging speed are known up to that date. For these wells a correction was applied to the raw data to account for the logging error. The

properties of the used sensor are presented in Table 2. Another five TD profiles were taken in a second campaign presented by Rogiers et al. (2015). A different temperature logger was used for which the characteristics are presented in Table 2. The stop-go measurement method by Harris and Chapman (2007) was followed similarly to the previous campaign.

The temperature probe used in this work was the *Star-Oddi's DST-mili TD* [https://www.star-oddi.com/products/data-loggers/depth-sensor-water-level-data-logger-recorder-milli-TD]. This probe is typically used in TD logging by tagging fish

in migration and behaviour studies. It was selected due to its low diameter making it ideal to overcome a sort of 'bottle neck' observed in a previous campaign in boreholes located in the ON-MOL-2 site. The dimensions of this probe are presented in Table 2 summarizing the temperature sensors used in this and in previous works. The measuring method described by Bense et al. (2017) and Kurylyk et al. (2018) was followed. In this work, 15 groundwater wells were measured, to extend the number of observations spread across the Nete catchment. Finally, a total of 35 TD profiles distributed in time and space across the

Nete catchment are used as observations to constrain the heat-transport model.

#### 3.1.2 Air and soil temperature time series

Several meteorological stations exist within the region. Only three provide measured values of air (at 1.75 meters above surface) and soil temperature within or nearby the study area: the Herentals, Overpelt and Eindhoven stations. Data from the two stations in the Flemish region, belonging to the Flanders Environment Agency (VMM) [https://www.vmm.be/], were

accessed through waterinfo.be [https://www.waterinfo.be/]. Data from Eindhoven was obtained through the Royal Netherlands Meteorological Institute (KNMI) [https://www.knmi.nl/home]. Temperature-time series for the data gathered for these stations from 01/01/2003 to 31/10/2016 is presented in Figure 3.



### 3.2.3 Remote sensing data

Remote sensing data play an important role in the development of validated multi-scale Earth system models (Fick and
Hijmans, 2017; Hazaymeh and Hassan, 2015; Tomlinson et al., 2011). Several satellite missions launched by NASA and ESA
(European Space Agency) collect data from the Earth's crust up to the atmosphere on temperature, gravity variations, landscape
characteristics, etc. Missions such as Landsat, ASTER, MODIS and Copernicus collect *i.a.* land surface temperature (LST)
values which can be freely retrieved from their online platforms. Landsat images, being the highest resolution with a daily
frequency, are unfortunately only available for the US. Both, Copernicus and MODIS missions gather global LST values.
However, MODIS gathers 1 km resolution images whereas Copernicus images have a 5 km resolution. For this reason, LST
images were retrieved from NASA's MODIS (Moderate resolution Imaging Spectro-radiometer) for the region, with 8-day
averages from 2001 to 2019. These were upscaled to monthly means for our purposes. The resulting raster stack was sampled
for the 2001-2019 monthly period in 2 locations, corresponding to the VMM weather stations of Herentals and Overpelt. These
results are also included in Figure 3. From this sampling, it is clear that the raster LST images overestimate the peak
temperature values, mostly during the summer months. Reinart and Reinhold (2008) discuss this effect on peak values,
similarly, by comparing MODIS retrieved temperature values and in situ measurements. They attribute this overestimation to
the so called "skin effect", since satellite radiometers are only able to retrieve skin temperatures (Reinart and Reinhold, 2008)
which are influenced by several other components i.e. prevailing wind speed, and time and conditions on the day the
measurement is taken (Fick and Hijmans, 2017; Tomlinson et al., 2011; Wan et al., 2002). A simple linear regression model
was fitted to correct the LST raster stack in relation to in situ measurements from the weather stations ($LST_{corr} = 0.6022$ LST
+ 3.162; $R^2 = 0.8$, standard errors: slope, 0.025; intercept, 0.46), see Figure 4. Although there seems to be some seasonal
hysteresis in the relation between the two, and one could potentially use *e.g.* the month as an additional regressor, we
considered this to be sufficiently accurate for the current work. This correction was then applied to every LST raster (i.e. every
month from 2001 to 2019). An example of the corrected monthly images for year 2001 is shown in Figure 5. From the corrected
monthly images it is possible to observe temporal but also the spatial distribution across the catchment.

### 3.1.4 Holocene paleo-temperature reconstruction

To advance the conceptualization of the transport model and to improve the initial boundary condition for the top of the model
(i.e. temperature values), a long-term temperature-time input curve was built using data from several authors. The data was
retrieved from the paleoclimatology data library of the National Center for Environmental Information (NOAA)
[https://www.ncdc.noaa.gov/data-access/paleoclimatology-data]). Data from D'Arrigo et al. (2005, 2006), Casty et al. (2007),
Buntgen et al. (2011), Tingley and Huybers (2013), Esper et al. (2014), Luterbacher et al. (2016), Langevin et al. (2017),
Marsicek et al. (2018), Ljungqvist et al. (2019) and Glaser and Riemann (2009) were used for the estimation of the temperature-
time input curve. The data from these authors present paleo-temperature reconstruction values for the Holocene in Europe, in
gridded format with coordinates for reference. Data from Mann et al. (2009) and Mann (2002) were used for comparison, since



190 they represent a general and internationally accepted Northern-Hemisphere temperature-time curve. The values from these
authors were taken from the gridded value or values that contained Flanders within its/their boundaries. The obtained values
from each author curve were then scaled to the relative Belgian average temperature for the period 1961-1990 derived from
Jones et al. (1999) and Jones et al. (2012). These temperature-time curves are shown in Figure 6. Inverse distance weighting
(IDW, $1/d^{0.1}$) was performed on the data locations with reference to the center of the groundwater model domain. The resulting

195 weights were used to determine an IDW average time series, for which yearly (IDW - 1y) average is displayed in Figure 7.
Recursive partitioning segmentation (RPS) was done using an implementation of the Lavielle (1999) and Lavielle (2005)
method. This method was applied to different sections of the IDW averaged temperature-time curve, aiming to obtain a
segmentation from -8500 to 2000 (CE, common era) using average step sizes of 1 kyr (-8500 to -500), 100 yr (-501 to 1500),
10 yr (1501 to 1950) and 1 yr (1951 to 2000). Monthly values for the last 19 years from remote sensing land surface temperature

200 data were included at the end of the segmentation (see Figure 7). A total of 90 time-steps are then included to the steady-state
solution for determination of initial temperature aiming to improve its efficiency.

**3.2 Modelling framework**

For the current work, the Neogene aquifer model (NAM), a steady-state groundwater flow model constructed by Casillas-
Trasvina et al., (accepted for publication, 2021) using MODFLOW-2005 (Harbaugh, 2005) is used. A heat transport model by

205 Rogiers et al. (2015) has been updated using MT3D-USGS (Bedekar et al., 2016) and is used for simulations. The models
were developed and post-processed using the RMODFLOW (Rogiers, 2015b, 2016a) and RMT3DMS (Rogiers, 2015b, 2016b)
packages for R (R Core Team, 2020).

**3.2.1 Conceptual model**

The NAM by Casillas-Trasvina et al., (accepted for publication, 2021) has a lateral boundary that coincides with the catchment

210 boundaries of the Kleine and Grote Nete rivers. Similar to Gedeon (2008), the catchment is assumed to be laterally isolated,
so no groundwater flow across the lateral boundary occurs. The top boundary is put at the ground surface elevation whilst the
bottom boundary coincides with the top of the Boom Clay formation. The Neogene aquifer becomes deeper in northeastern
direction, from the southwest corner where the Boom Clay is present at the ground surface. The groundwater flow in the
Neogene aquifer is driven mainly by surface hydrological features (i.e. recharge and rivers) creating several local flow systems,

215 with local influences of abstraction wells. It is assumed to be in dynamic equilibrium, with no long-term trends in groundwater
flow, which allows for a steady-state simulation (Gedeon, 2008). Casillas-Trasvina et al., (accepted for publication, 2021)
included the conceptual model consideration that fault planes have hydraulic properties distinct from the surrounding
sediments, and do not penetrate up to the surface, as they are buried by at least a thin Quaternary cover. Additionally, these
planes are assumed to act as barriers if they behave in a significantly different way than the surrounding sediments, affecting

220 the horizontal component of the groundwater flow. Together with their buried nature, this causes upward flow at the upgradient
side of the fault. On the downgradient side of the fault, the vertical flow direction may also be affected turning into a more





parallel direction in respect to the fault. Finally, groundwater may be overflowing above the fault plane if it acts as a strong flow barrier, given the fact that the fault is buried, creating a groundwater level step (head difference) as observed at the ON-MOL-2 site.

**3.2.2 Groundwater flow model**

The structural updates into the NAM for heat-transport follow the latest hydrostratigraphic 3D model for Flanders (H3D) by Deckers et al. (2019). Similarly as the NAM by Casillas-Trasvina et al., (accepted for publication, 2021), it assumes 9 hydrostratigraphic units: non-tabular Quaternary (Quaternary cover above the 'Kempen' aquifer system, Weelde, Malle, Merksplas, Mol, Lillo, Kasterlee, Diest, and Berchem & Voort of which the geometry is based on Deckers et al. (2019) The

model domain is discretized in 49 vertical layers thinning out closer to the surface to ensure smaller modelling cells close to surface hydrological features were groundwater gradients are the highest. The bottom of layers 1 and 2 were assigned equal 50% and 30% of the ground surface elevation. The bottom of layer 3 was set equal to 0 m a.s.l. (TAW). From layer 4 to layer 9, layer thicknesses of 5 m are used. From layer 10 to 49, the thickness used is 10 m. The modelled area was discretized into a regular model grid of 96 rows and 146 columns resulting in cells with dimensions of 400 m × 400 m. A total of 23 faults

were simulated with the horizontal flow barrier (HFB) package (Harbaugh et al., 2005) starting from the top of the second numerical layer (from around 12 to 18 meters below surface) to the bottom of the modelled domain, given that the faults do not present a clear surface expression (Verbeeck et al., 2017). Rivers, lakes, canals and abstraction wells are defined in the groundwater flow model. The groundwater abstractions range from a few $m^3/d$ to more than 300 $m^3/d$. Data from several sources were used to define these parameters including the Flemish hydrographic atlas ("Vlaamse Hydrografische Atlas",

VMM 2017) and the IGN/NGI dataset (IGN/NGI 2017). Spatially distributed recharge is implemented with values obtained from DiCiacca (2020) which are derived from vegetation cover, soil texture and spatial input layers of depth to the groundwater table based on hydraulic head observations. A scaling factor was used during the model inversion for the calibration of the recharge initial value. A total of 1393 averages from hydraulic head time series are used in the NAM. These observations were obtained from the piezometric network monitored and maintained by SCK CEN for ONDRAF/NIRAS, and from the

subsurface database for Flanders (Databank Ondergrond Vlaanderen; DOV).. The model performance of the NAM model had a RMSE of 0.70 meters accounting for a total head loss of 46.5 meters. Figure 8 (a, b and c) show results from the groundwater flow model: a) the hydraulic head distribution over the Nete catchment; b) the scatter plot of simulated equivalent vs observed hydraulic head observations; and c) a cross section (B-B') with arrows indicating the flow magnitude results from the groundwater flow model per model cell. Figure 9 shows a cross section with derived Peclet numbers ($P_e$) following Freeze and

Cherry (1979) ($P_e = \frac{V_e d}{D_e}$), relating the rate of flow advection to the diffusion. In here, a value of $2x10^{-9}$ $m^2/s$ for the diffusion coefficient ($D_e$) is used corresponding to the tritium diffusion coefficient in pure water (Gedeon, 2015), and a value of L = 200 m, corresponding to a rough average distance from the bottom of the aquifer to the surface. It is notable that $P_e>>1$ in the formations above Berchem and Voort Sands, thus being more advection-dispersion dominated areas. On the other hand, in the



deepest areas of the aquifer (i.e. Berchem and Voort sands), Pe ≤1 therefore diffusion becomes a more dominant transport
mechanism. A more detailed description of the NAM structure, parameters and results is presented in Casillas-Trasvina et al.,
(accepted for publication, 2021).

### 3.2.3 Heat-transport model development

*Numerical modelling*

In Liu et al. (2019) and Hecht-Méndez et al. (2010) it is demonstrated that MT3DMS is able to simulate heat transport when
temperature variations are limited, through the analogy with solute transport. If large variations in temperature are to be
modelled (i.e. aquifer thermal energy storage (ATES), ground source heat pump (GSHP) with large ranges) a model that
accounts for variable density and viscosity terms (*e.g.* SEAWAT (Langevin et al., 2008a), SUTRA (Voss and Provost, 2010),
FEFLOW (Trefry and Muffels, 2007)) should be applied, but in the current setting, we do not consider this to be relevant. The
heat and solute transport equations formulated in analogous forms show how MT3DMS can be used for heat transport
simulations. For single species, the solute transport equation solved by MT3DMS (Langevin et al., 2008a) is as follows (Eq.
1):

$$\left[1 + \frac{\varphi_b K_d}{\theta}\right] \frac{\delta(\theta C)}{\delta_t} = \nabla * \left[\theta\left(D_m + \alpha\frac{q}{\theta}\right) * \nabla C\right] - \nabla(qC) - q_s' C_s \tag{1}$$

where: $\varphi_b$ = Bulk density (mass of the solids divided by the total volume) [ML⁻³]; $K_d$ = Distribution coefficient of solute
[L³M⁻¹]; $\theta$ = Porosity [-]; $C$ = Concentration of solute [ML⁻³]; $\nabla$= Vector differential operator; $D_m$ = Molecular diffusion
coefficient [L²T⁻¹]; $\alpha$ = Dispersivity tensor [L]; $q$ = Specific discharge vector [LT⁻¹]; $C_s$ = Source concentration of solute
[ML⁻³]; $q_s'$ = Source or sink of fluid [T⁻¹].

The analogous equation (Eq. 2) for the heat transport is defined as:

$$\left[1 + \frac{1-\theta}{\theta}\frac{\varphi_s C_{p\,solid}}{\varphi C_{p\,fluid}}\right] \frac{\delta(\theta T)}{\delta_t} = \nabla * \left[\theta\left(\frac{K_{T\,bulk}}{\theta\varphi c\,fluid} + \alpha\frac{q}{\theta}\right) * \nabla T\right] - \nabla(qT) - q_s' T_s \tag{2}$$

where: $T$ = Temperature [°C]; $ps$ = Density of the solid [ML⁻³]; $\varphi C_{p\,fluid}$ = Specific heat capacity of the fluid [L²T⁻²°C⁻¹];
$\varphi C_{p\,solid}$ = Specific heat capacity of the solid [L²T⁻²°C⁻¹]; $K_{T\,bulk}$ = Bulk thermal conductivity of the aquifer [ML³T⁻²°C⁻¹];
$T_s$ = Source temperature [°C].

When the more recent MT3D-USGS code (Bedekar et al., 2016) is used to simulate heat transport, these equivalent transport
parameters are defined. The heat dispersion is the same as solute dispersion, which is determined by longitudinal and
transversal dispersivities. For a more detailed description the reader is referred to Anderson 2005; Langevin et al. 2008b;
Zheng 2010; Bedekar et al. 2016; des Tombe et al. 2018; Liu et al. 2019.





### *Modelling set-up*

A heat-transport model of the original NAM (Gedeon, 2008) using MT3DMS (Zheng, 2010) has been constructed by Rogiers
(2015b). In this model, the top boundary condition uses a yearly average value for the temperature imposed on the ground
surface. The transport simulation covered a period of 10,519 years, with yearly time-steps, covering almost the entire Holocene
period. The set temperature as boundary condition at the surface accounted for the topographical influence, and land cover
classes, based on data presented by Leterme and Mallants (2012). A heat flow density of 0.06 Wm$^{-2}$ was set to the bottom
boundary condition, value in accordance to the average heat-flow density reported by Pollack et al. (1993) for Cenozoic and
Mesozoic sedimentary and metamorphic crust (Rogiers et al., 2015). The specific heat capacity, $\varphi C_{p\ fluid}$, and density, $pw$, of
the fluid (groundwater) were set to 4193 Jkg$^{-1}$K$^{-1}$ and 999.7 kgm$^{-3}$, respectively  (Rogiers et al., 2015). Material properties
defined in the model were a) total porosity ($\theta$) b) specific heat capacity ($\varphi C_{p\ fluid}$), c) density of the solid ($ps$), d) thermal
conductivity ($\lambda m$ ) and dispersivity ($\alpha$).

In this work, similarly to Rogiers et al. (2015), we based the material properties on reported literature values: Hoes et al.
(2005); Beerten et al. (2010); Chen et al. (2011); Govaerts et al. (2011) and the WTCB methodology reported in van Lysebetten
et al. (2013). These values and a summary of the hydrologic properties required to parametrize the MT3D-USGS model used
in the simulations are shown in Table 3. Note that the potential impact of geothermal heat flow of an increasing temperature
with depth on viscosity and density of the fluid are not considered. This assumption is reasonable as we consider a not so deep
groundwater flow system (<500 m) where these temperature effects will be of minor importance (Bense and Person, 2006;
Person et al., 1996), and where emphasis is not put on surface water/groundwater interactions where the impact on viscosity
derived from temperature gradients may be considerable (Engelhardt et al., 2013; Kurtz et al., 2014; Liu et al., 2019; des
Tombe et al., 2018).

Building on the previous work, the latest version of the heat-transport model built by Rogiers et al. (2015) has been updated
from MT3DMS (Zheng, 2010) to MT3D-USGS (Bedekar et al., 2016) following the new hydrostratigraphy and grid
discretization used for the updated NAM groundwater flow model by Casillas-Trasvina et al., (accepted for publication,  2021)**.**
The dispersivity was based on a scale of 200 m, which is a rough measure for the depth of the aquifer, and thus the travel
distance of heat.The linear dispersivity-scale relationship was derived from literature data, as presented by Rogiers et al. (2013).
For simplicity, the dispersivity is considered to be constant, with longitudinal dispersivity $\alpha_L$ equal to 6 m, and the ratios of
transverse and vertical transverse dispersivity to $\alpha_L$ equal to 0.1 and 0.01 respectively.


### *Top boundary condition*

As first modeling step, a steady-state finite-difference solution was obtained using temperature from year -8500 as top
boundary condition. This was used as initial temperatures for transient simulations. The transient model is run for 10,519 years,
from -8500 to 2019. Every single time step is derived as explained above in section 3.1. A yearly average value from the LST
raster stack was determined to be used as spatial distribution for the Holocene paleo-temperature distribution implemented in

the model as top boundary condition. The temperature spatial distribution was derived from satellite data and used directly in the monthly part of the transient simulation. The values for every 1-km LST raster were downscaled in the center of each active model cell (MODFLOW model) with a 400m x 400m resolution. A total of 316 rasters were produced from years -8500 to 2000, and one for each month, from January 2001 to October 2019. Each of these rasters represent a time-step for the

transient-transport simulation.

*Model calibration*

Automatic parameter optimization was implemented as technique for model inversion. Several algorithms were used for global model optimization such as the Standard Particle Swarm Optimization (spso11) (Clerc et al., 2012; Zambrano-Bigiarini et al.,

2013; Zambrano-Bigiarini and Rojas, 2013), and Differential Evolution (Mullen et al., 2011). Temperature gradients derived from each TD profile and hydraulic head values were used as calibration targets during the joint inversion procedure. Results from the spso11 algorithm were selected for minimizing the root-mean squared error (RMSE) despite requiring a slightly longer computational time.

**3.2.4 Modelling strategy**

*Model 1: Baseline*

The updated version of the groundwater flow model by Casillas-Trasvina et al., (accepted for publication, 2021) and the heat-transport model, including the Holocene temperature-time curve and the satellite data for the last 19 years, is defined here as the baseline model (Paleo RPS). The heat-transport simulation is jointly inverted with the groundwater flow model against

temperature gradients and hydraulic head observations, respectively. The results from this model are compared against modelling cases 2 and 3 to estimate the effect of thermal conduction and normal faults in the temperature distribution in the Neogene aquifer, across the Nete catchment. Additionally, a model without the effect of the paleo temperature including only the LST temperatures in the last 19 years (named monthly LST model), and a steady-state transport model are also evaluated in comparison to the defined baseline model.


*Model 2: Thermal conduction*

A model case accounting for only thermal conduction is performed to enable quantification of the effect of groundwater flow on the subsurface temperature distribution. The baseline simulation accounts for both advection of heat via groundwater flow and thermal conduction. For this second simulated model without groundwater flow, the same parameter set is used, but the

advection of heat is switched off. The difference between the first and second simulation temperature distributions then gives us the changes in temperature induced by the groundwater flow.

*Model 3: Heat-transport without faults*





An exploration of the temperature distribution in the aquifer without faults (i.e. no HFB included in the flow model by Casillas-
Trasvina et al., (accepted for publication, 2021) is also performed. For the heat-transport model, parameters are set the same
as for the baseline model. Both this case and the baseline model account for both advection of heat via groundwater flow and
thermal conduction. The difference between both these cases allows the quantification of the effect of faults on the temperature
distribution, and provides an idea on the parameter sensitivity of the hydraulic conductivity of such structures.

## 4 Results and discussion

### 4.1 Paleoclimate effect on temperature distribution

A scatter plot of the model simulated temperature gradient values vs the observed measurement is shown in Figure 10a, where
a comparison is made between the monthly LST model (see 3.2.4, model 1: baseline), and the baseline model of the current
work (Paleo RPS), separating the observations with a z value below and above -5 m a.s.l.. A steady state transport solution
result is also included for reference. From this scatter plot, both transient heat transport simulations have a bias towards lower
gradient values, indicative of flux magnitudes being potentially overestimated. Additionally, most of the simulated values
above the z = -5 m a.s.l. limit show large fluctuations. This may be explained by the imposed temperature value as the top
boundary condition (at the first layer of the model), the potentially heterogeneous properties of the materials near the surface
and the space-time discretization of the model (i.e. dimensions of the cells, time-step sizes). The spatial variations in the
imposed temperature (i.e. scaling of the thermal images/model cell size vs the scale where TD profiles were measured) together
with the local flows occurring near the surface of the model affected in very local scales (e.g. surface/groundwater exchanges)
participate in these fluctuations that seems to stabilize at around -5 m a.s.l. depth. At these shallow depths temperature
variations can drive fluctuations in heat transport related to the fluid viscosity (Hecht-Méndez et al., 2010; Liu et al., 2019; des
Tombe et al., 2018), limited to these few meters (~20 m) below the surface. Additionally, although yearly and monthly time-
steps were defined in the model, seasonal/diurnal temperature changes may also affect the propagation of heat into the
subsurface (Benz et al., 2018; Dong et al., 2018). Nevertheless, the simulated results present an acceptable compromise
between observed and simulated values. The paleo-temperature reconstruction model results are more clustered, less disperse
despite having some values spreading at a lower simulated equivalent gradient, in comparison with the LST transient model
(see Figure 10c), thus indicating that this simulation is overall being slightly more accurately representing the observed
situation (RMSE = 0.03 °C/m; 1.15 °C), though several inaccuracies are observed in deeper parts of the aquifer (Figure 10c ).

### 4.2 Current temperature distribution

The flow and transport parameter composite scaled sensitivities (CSS) with respect to hydraulic head and temperature gradients
respectively, are shown in Figure 11. It can be seen that the flow parameters have a much larger effect on the model results
than the temperature parameters implying their relevance to the produced results. In terms of temperature, the Diest Formation



thermal conductivity presents the highest sensitivity, given that most of the temperature profiles were taken at these depths. On the contrary, the Berchem Formation (Berchem & Voort Sands), not having a large number of observations, presents relatively high sensitivity as well. Given that at these depths the magnitude of groundwater flow is very low (around $1\times10^{-7}$-$1\times10^{-5}$ m/d), the transport of heat is less advective, and more driven by thermal conduction. Nevertheless, in general, flow model parameter sensitivities are far larger than those for heat-transport model given that thermal properties have less variation than the hydraulic ones (Saar, 2011).

The temperature distribution at the end of the calibrated heat-transport model simulation is shown in Figure 12. In Figure 12a, a cross-section B-B' is shown, indicated in Figure 12b.Figure 12a shows a distribution  that closely relates to groundwater flow as seen in Figure 8, indicating the influence of advection. At the east of the Rauw Fault colder water infiltrates, travels downwards and mixes with warmer groundwater from the bottom of the aquifer. Groundwater with a slightly higher temperature is transported upwards before penetrating the Rauw Fault, however with most groundwater flowing across the fault at around -150 m a.s.l.. Then, upstream of the Rauw Fault, a portion of the groundwater flowing upwards overflows above it to then continue its flow direction downwards mixing again with colder infiltrated water at the west of the Rauw Fault. Groundwater to the west of the Rauw Fault then gets mixed with colder and younger infiltrated water reducing its temperature along the approximate flow direction (i.e. to the west) until the end of the fault block, to then continue flowing laterally to the west merging with another more local flow system. This can be seen from Figure 12b, which displays the model layer 10, where groundwater temperature to the west of the Rauw Fault zone is higher than to the east. Groundwater in this zone is largely influenced by the surface water features carrying groundwater from deeper in the aquifer that has already crossed the Rauw Fault. Groundwater to the west of the Rauw Fault seems to have a lower temperature given that mixing with warmer groundwater occurs deeper, near the bottom of the aquifer.

In the shallower parts of the aquifer (z > -100 m a.s.l.; Figure 12a) some 'isolated zones' with lower temperatures can be identified in the vicinity of other faults (i.e. from east to west: 'HFB_094', 'HFB_089', 'HFB_129'; represented as blue dashed lines). These 'isolated zones' correspond to the groundwater flow distribution producing local flow systems in the area formed by the relatively large density of rivers and canals. Even though the faults in the vicinity have a relatively low sensitivity (see Figure 11), these might still have an effect on the distribution of temperature. In their review, Irvine et al. (2012) mention that although thermal properties might have low variations, as stated by Saar (2011), these variations might have a reasonable effect in the overall temperature distribution in sedimentary environments (Chang and Yeh, 2012; Constantz et al., 2003; Hidalgo et al., 2009), such as the Neogene aquifer. On the other hand, in the deeper parts of the aquifer (near the bottom of the aquifer, Figure 12a), higher temperatures are present (close to the main source of heat to the aquifer). At these depths, groundwater fluxes are low (~$1\times10^{-7}$ m/d), which would indicate that the heat transport here would be mostly conduction-driven. The discharge zone of the aquifer downstream to the Rauw Fault location seems to have some influence from the surface water network driving deep groundwater to the surface. Groundwater abstractions in this zone seem to have an impact, added to the effect of the surface water network, driving heat from deeper parts of the aquifer. into more superficial groundwater flow systems.





At the eastern side of the Rauw Fault, colder temperatures are found from infiltrating water  up to  z ~ -150 m a.s.l.. At
approximately this elevation, infiltrating colder water mixes with warmer groundwater from the bottom of the aquifer (Figure
12a, green arrow), delineating an area where mixing of younger and older groundwater seems to occur that appears to be
defined by the flow velocities characteristic of the hydrostratigraphic formations (z ~ -200 m a.s.l., at around the limit of the
Diest Formation with Berchem & Voort Formations). On the other hand, at the western side the Rauw Fault zone, an
intercalation of the local flow systems is seen, following the east to west flow direction (Figure 12a, cyan arrows, from east to
west), as i) infiltration area with colder waters, ii) upwards groundwater flow driven by the surface water network and
groundwater abstractions, iii) infiltration area with cold water, and iv) upwards groundwater flow. In the section where the
aquifer becomes thinner at around 100 meters thick  (around the center of the aquifer) towards the west, several even narrower
local flow systems occur (as seen in Figure 8c and Figure 12). Although, these patterns are difficult to discern in the current
figure, the majority of the temperature anomalies indicate upward fluxes, which can be explained by the relatively larger
density of rivers and drains in the area.

Selected locations along the approximate flow direction where TD profiles were measured are shown in Figure 13a. Their
measured and calculated TD profiles, for the three modelled cases (i.e. Paleo RPS, only conduction and without faults), are
shown in Figure 13b (all measured TD profiles are shown in Annex 1). From east to west, the measured sites are R-13d (to the
east of the Rauw Fault), R-54c, R-11c, R-15f, R-2c, R-34c, R-1b, R-51c and R-43c (to the west of the Rauw Fault). All
measurements show a C-shaped profile, an indication of transient conditions at the surface (Anderson, 2005; Bense et al.,
2017). From Figure 13b the simulated and observed temperature values show that the closer the TD profiles were taken to the
fault (R-54c being the closest one), the larger the absolute temperature error is for either of the three simulated cases. The
simulated temperature at R-54c is lower than observed, which can be explained by its close proximity to the Rauw Fault (of
around 10 meters apart) and thus being largely influenced by the flux across it which might be overestimated, as indicated by
the lower RMSE by the simulation case accounting only thermal conduction. The results from the simulation only accounting
for thermal conduction show the overestimation of the groundwater fluxes in few locations, mostly those located from the
center to the east of the studied area. Although the disparity in the model performance is relatively small, it shows the influence
that the dense network of rivers and lakes in this zone has on the temperature distribution. Towards the west, where the density
of rivers is lower and the aquifer becomes thinner, these differences become nearly negligible. The complete temperature-
depth profile time series simulated by the paleo-transport model is included (gray lines), indicating the range of potential
temperature values that this model with these specific boundary conditions and heat source can achieve. These paleo-
temperature simulated values, starting low, seem to stabilize after several transport-steps, until achieving an apparent relatively
steady condition in the deeper parts of the aquifer (z < -150 m a.s.l.). Similarly as presented by Dong et al. (2018) and Benz et
al. (2018), long-term temperature variations in the top boundary condition are potentially driven by external factors such as
climate change or changes in land use/cover. These factors seem to have an influence in temperature observations at depths of
around -50 m a.s.l. or shallower, according to Figure13b. In Figure 13a the complexity of the river and canal network is
visualized, and several lakes close to the measured sites can be observed. These lakes might explain anomalies occurring





relatively close to the surface, but given that groundwater is mainly driven by the surface water network, they represent a likely driver for these shallow variations, potentially driving deeper/warmer waters upwards. Given that the simulated and measured

temperature gradients are in fair agreement, the surface water network might have some influence on the temperature top boundary condition for which LST temperature images and the paleo-temperature time curve might not be accounting for.

### 4.3 Temperature distribution cases

A comparison between the simulated temperature fields by the baseline (paleo-temperature reconstruction) model versus a) thermal conduction only (no advection), and b) without faults as HFB, is shown in Figure 14, and described here.

#### 4.3.1 Thermal conduction case

*Local flow systems patterns*

Removing the advection transport mechanism would allow temperature to be freely conducted without being transported by the groundwater flow. It is clear that advective groundwater flow has a large impact on the simulated temperature fields and thus it is one of the main mechanisms for heat transport in the Neogene aquifer (see Figure 9). Given that advection is not

being considered (no temperature convection), Figure 14b clearly shows a temperature pattern that mimics the surface water network. This strongly supports the statement that heat transport in the aquifer may indeed be mainly driven by the advection in local flow systems, where temperatures increase near rivers and streams at the surface by deep groundwater discharge, and by large advection related to groundwater abstractions. The interfluves are shown to be where most negative values are located, clearly indicative of higher temperatures in the purely conductive model. These areas are indicative of downward flows serving

as recharge areas feeding the local flow systems which in turn are being discharged in more downstream areas by the surface water drainage effect. The effect of these local flow systems seems to depend on the depth of the aquifer. In the north-eastern part of the study area, where the aquifer is thickest, the largest effects are present, with respect to the pure thermal conduction scenario, while in the south-west, where the aquifer is thin, there is in fact very little difference.

*Depth of the aquifer vs local flow system*

Groundwater on the downstream side of the Rauw Fault, in the upper part of the aquifer at around z = -50 m a.s.l. (Figure 14a), can be seen as warmer when advection is considered due to the Rauw Fault overflowing effect, as well as areas below rivers. However, in the deeper parts of the aquifer, higher groundwater temperatures are shown for the thermal conduction case, suggesting that conduction may potentially be the dominant mechanism of heat transport (due to the lower hydraulic

conductivity, see above). Large temperature differences can be found in several locations where abstraction wells are located. The withdrawals made by these abstraction wells drive groundwater to flow upwards at several locations (see Figure 8c) hence transporting heat at various depths from the bottom of the aquifer to shallower levels. The effect of the local groundwater flow systems driving heat from the bottom of the aquifer at several locations along the apparent flow direction is relatively widespread across the aquifer (as seen in Figure 14a). However, although groundwater flow magnitudes are the lowest (~1x10⁻



[7] m/d) at the bottom of the aquifer, groundwater abstractions represent an important driver for the upwelling of deep warmer
groundwater at these depths. By this groundwater upwelling, parcels of groundwater flow upwards in the surroundings of the
abstraction well, which are then spread over the aquifer by the natural circulation of local groundwater flow systems.

### 4.3.2 Heat-transport without faults

In Figure 14c, the delta temperature distribution of the baseline model minus the heat-transport without faults model is shown.
It indicates that a considerable variation on the temperature to the western side of the Rauw Fault zone would occur.
Groundwater would flow freely and without the need to find a way across, either by slowing down or being forced upward
east of the Rauw Fault. Groundwater with a higher temperature at the bottom of the aquifer would be driven in the upwards
and downstream directions, having a relatively broad spread of heat. Adjacent to the Rauw Fault on the eastern side, at around
$z \sim -100$ m a.s.l., a higher temperature zone can be observed. This indicates the approximate depths where groundwater begins
to flow upwards and heat is being transported across the fault in the top last few meters, crossing, and creating the occurrence
of higher temperatures adjacent to the Rauw Fault at the western side. These relatively higher variations in temperature occur
in the vicinity of the Rauw Fault (Figure 14c). The results suggest that further downgradient, a much lower or even negligible
impact exists (Figure 14d).

## 5 Conclusions and recommendations

Our study provides highly detailed spatial and temporal temperature data (i.e. the LST raster stack and a paleo-temperature
time series) for the implementation of boundary conditions for reconstructing the past and current temperature distribution
with the use of a transient heat-transport model. Limitations that are typically associated with applying a specified temperature
across the top of the model domain are tackled here by constructing a Holocene temperature-time curve to define a surface
temperature boundary for the subsurface spatially distributed temperature to reduce the uncertainty in the temperatures at the
water table. Additionally, we collected several state variable observations (*i.e.* TD profiles) from literature and executed
different temperature logs ourselves as well, for the purpose of testing the process model, and performing calibration. This
study builds up on the previous efforts providing an upgraded example of the combined impact of conductive and advective
phenomena associated to paleoclimatic fluctuations to characterize the temperature distributions observed in the Neogene
aquifer.

Long-term transient heat transport simulations were done using a steady-state groundwater flow field, and calibrated on the
basis of TD profile measurements in the Neogene aquifer across the Nete catchment. The model provides an acceptable
representation of the groundwater system since all of the simulated TD profiles are within 2 °C (0.03 °C/m) of the simulated
(with a maximum difference of 1.7 °C). Seventy percent of the simulated values TD profiles are within 1 °C (0.015 °C/m) of
the observed values and 50 percent of them are within 0.5 °C. In some areas the modeled behavior deviates from the observed





temperature-depth values, this is regarded as being due to groundwater flow over/under estimations which drives the advection of heat in addition to conduction. Although some of these deviations may be relatively considerable in a few areas, they are indicators of local processes and/or drivers that may be isolated or combined, that the current models do not account for. In general, absolute temperatures at very shallow depths (Z >-5) and in the deepest parts of the aquifer are more difficult to

reproduce due to e.g. the imperfectness of the top boundary condition and/or local heterogeneities, however the temperature gradients have shown a good fit for most of the aquifer central and deep areas.

Because groundwater temperatures are highly affected by the magnitude of groundwater flow, it is clear that advection and convection play a major role as mechanisms of heat transport in the Neogene aquifer ($P_e \gg 1$). This behaviour changes with depth, where deeper in the aquifer groundwater fluxes are lower, and thermal conduction seemingly becomes dominant ($P_e \leq$

1). Assumptions have been made on particularly i) homogeneous thermal properties, and ii) homogeneous heat flux source. By these assumptions we neglect the variability of the presumably heterogeneous thermal properties (mainly in clay-rich formations) and heat source flux which may potentially introduce local variations in the temperature distribution where strong heterogeneity areas occur. Despite this, and given that advection/convection is seemingly the main mechanism of heat transport in the system, these uncertainties might only be relevant in some parts of the system: for instance i) at the surface derived by

the hysteresis produced by the LST vs soil temperature measurements; ii) where low groundwater fluxes occur, for instance, right at the bottom of the aquifer; iii) and possibly in clay-rich formations, such as the Kasterlee Formation. Future research is therefore suggested to be focused on these topics. Nonetheless, given the acceptable agreement between observed and simulated temperature values and gradients, and the distribution of temperature across the aquifer, a good approximation to the 'real' thermal properties and heat flux source has been implemented.

With the developed baseline model we evaluated different conceptual models to quantify the separate effects of advection, and faults acting as a barrier, and we see:

    i.    how meaningful groundwater flow is in spreading heat across the whole model domain, accentuating the drainage effect of the surface water network, corresponding local flow systems, and groundwater withdrawals; and

    ii.    the role of subsurface features, specifically the Rauw Fault in this particular case, acting as barriers/conduits

535            disturbing temperature distribution, although mainly locally.

These aspects are both of utmost importance particularly in the context of geological disposal of radioactive waste, but also advance our process understanding in view of the use of temperature observations as calibration targets, and their implementation in a joint calibration. For future investigations near the surface, a refined grid cell size (e.g. few to a couple tens of meters) would improve simulation accuracy, especially in areas of hydrogeological or thermal complexity, such as in

the surroundings of the Rauw Fault. Currently, the relatively coarse grid cells generalize relevant local-scale complexities (i.e. surface water/groundwater exchanges) that can potentially affect local flow paths and temperature gradients.

All these temperature variations serve as a proxy of the influence of advective transport and the role of faults as barriers or conduits, such as the Rauw Fault, and suggest that solutes released from the Boom Clay might be affected in similar ways. While our study used temperature variations to characterize subsurface transport at the catchment scale and in the vicinity of

the Rauw Fault zone, in the future other environmental tracers will be evaluated. These unconventional state variable observations are to be implemented for further examination of the groundwater flow and age distribution of the Neogene aquifer, and to be used in a joint inversion procedure to constraint numerical models for groundwater flow and solute transport.

**Author Contribution**

ACT, BR and KB designed the experiments, collected and processed data in different field campaigns. ACT carried the
modelling experiments and performed simulations with contributions from BR. BR, KB and KW provided supervision during the research period. LW (ONDRAF/NIRAS) provided part of the financial support and resources in this work. ACT prepared the manuscript with contributions to writing, reviewing and editing from all co-authors.

**Data availability**

Data available on request from the authors

**Competing interest**

The authors declare that they have no conflict of interest.

**Acknowledgements**

This work is performed in co-operation with, and with the financial support of ONDRAF/NIRAS, the Belgian Agency for Radioactive Waste and Fissile Materials, and SCK CEN Academy by SCK CEN, the Belgian nuclear research center, as part
of the programme on geological disposal of high-level/long-lived radioactive waste that is carried out by ONDRAF/NIRAS. The authors would like to acknowledge Dr. Victor Bense for his advice and support during the first temperature-depth measurement campaign of 2019.

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

**Tables**

**Table 1 Compiled historical TD profile data**

| Campaign | Year | TD profiles measured |
|---|---|---|
| Belgian Geological Survey | 1988 | 5 |
| SCK CEN | 1995 | 3 |
| SCK CEN | 1996 | 1 |
| PHYMOL Project | 1997 | 2 |
| Exploratory temperature logs (Rogiers, 2014) | 2012 | 4 |
| Rogiers et al. (2015) | 2015 | 5 |
| Current work | 2019 | 15 |

**Table 2 Summary of temperature sensors used for different campaigns.**

| Feature | Rogiers (2014) | Rogiers et al. (2015) | Current work |
|---|---|---|---|
| Probe model | Schlumberger mini-diver | Solinst levelogger edge (M200) | Star-Oddi DST mili TD |
| Dimensions (mm) | Ø 22x90 | Ø 22x159 | Ø 13x39.4 |
| Weight (gr) | 70 | 129 | 9.2 |
| Range (°C) | -20 to 80 | -20 to 80 | 0 to 85 |
| Accuracy ( °C) | ± 0.1 | ± 0.5 | ± 0.1 |



| Resolution (°C ) | 0.01 | 0.003 | 0.03 |
|---|---|---|---|


**Table 3 Overview of the hydrogeological unit properties used for the updated version of the NAM heat-transport model in MT3D-USGS**

Parameter names: [1]n = porosity, [2]Cs = specific heat capacity, [3]ρs = density of the solid phase, [4]λm = thermal conductivity, [5]$\alpha_L$

= dispersivity.

| Model Units | [1]n (%) | [2]Cs (J/kgK) | [3]ρs (gcm$^{-3}$) | [4]λm (W/mK) | [5]$\alpha_L$ (m) |
|---|---|---|---|---|---|
| non-tabular Quaternary | 35.9 | 800 | 2.62 | 1.5 | 6 |
| Weelde | 35.9 | 800 | 2.66 | 2.47 | 6 |
| Malle | 40.2 | 800 | 2.66 | 2.47 | 6 |
| Merksplas | 40.2 | 800 | 2.66 | 2.47 | 6 |
| Lillo | 40.2 | 800 | 2.66 | 2.47 | 6 |
| Mol | 40.2 | 800 | 2.66 | 2.47 | 6 |
| Kasterlee Clay | 39.7 | 800 | 2.65 | 2.1 | 6 |
| Diest | 41.1 | 800 | 2.75 | 2.1 | 6 |
| Berchem & Voort | 39.5 | 800 | 2.83 | 2.1 | 6 |
| Boom Clay | 37.3 | 769 | 2.65 | 1.31 | 6 |





## Figures

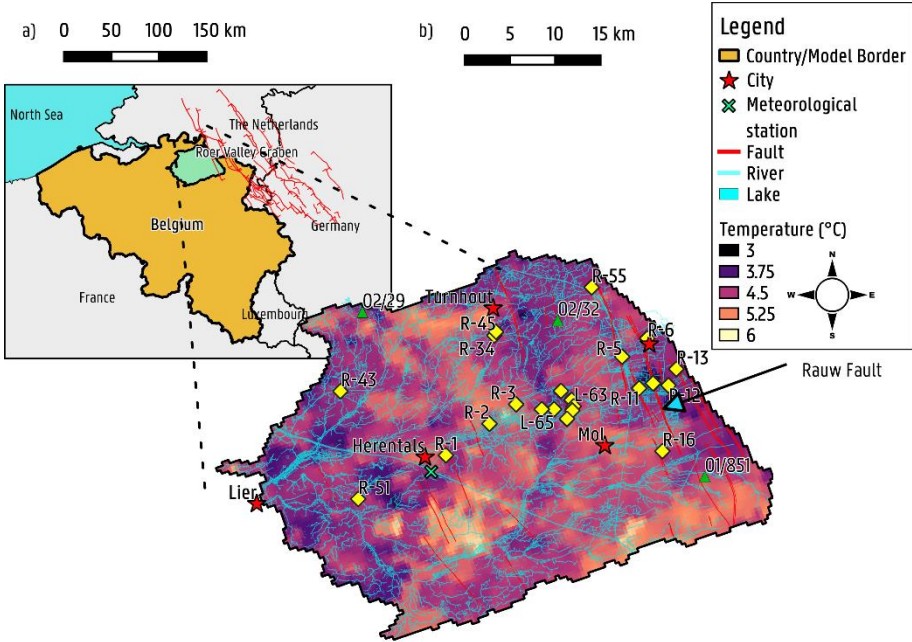


**Figure 1: Geographical location of a) The Nete catchment within Belgium with indication of the faults in the Roer Valley Rift System from Vanneste et al. (2013), b) the land surface temperature averaged for January 2001 derived from satellite data (MODIS) for of the study area within the Nete catchment.**

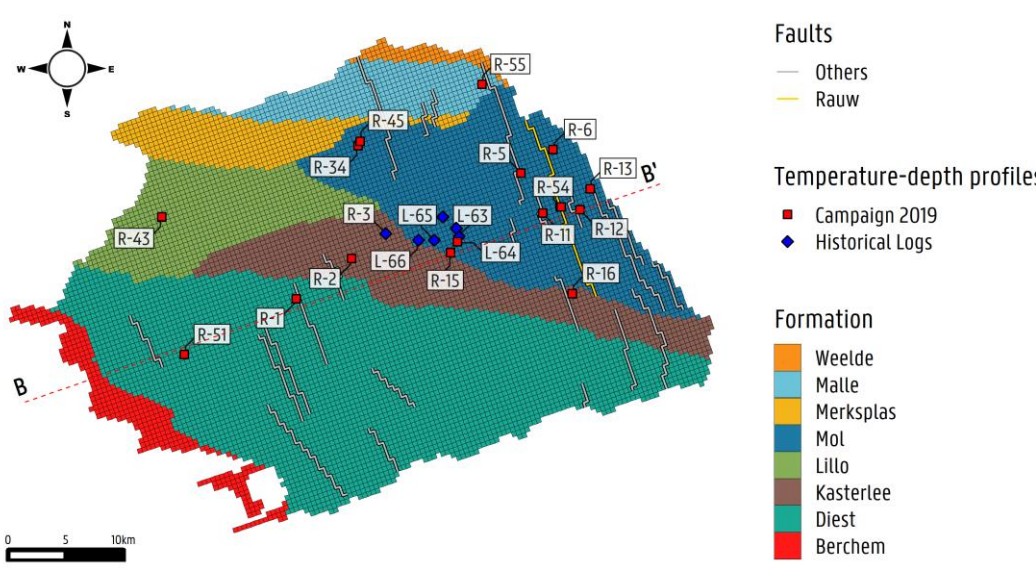


**Figure 2 Plan view of the study area as discretized in the second layer of the numerical model. It indicates faults (emphasis on the highlighted Rauw Fault), cross sections, temperature-depth profile locations, and modelled formations.**





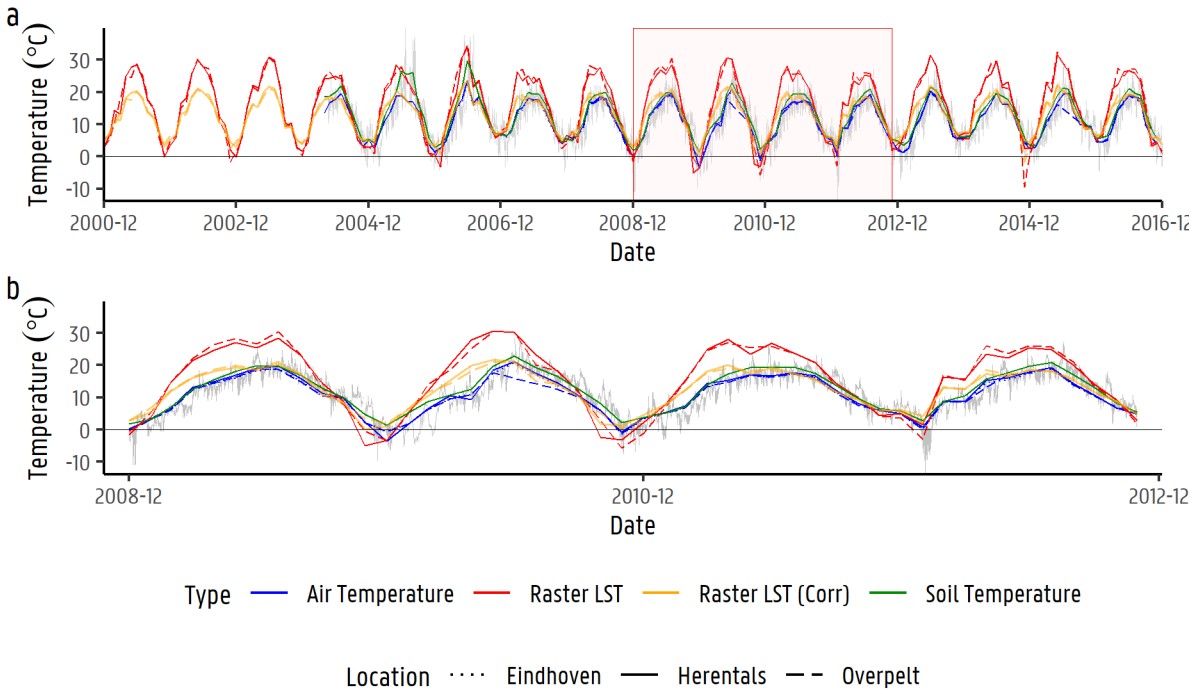

**Figure 3 Temperature-time series of air and soil measured in different stations within and nearby the Nete catchment.**
**Corresponding raw and corrected average land surface temperature (LST) based on remote sensing data are presented as well.**

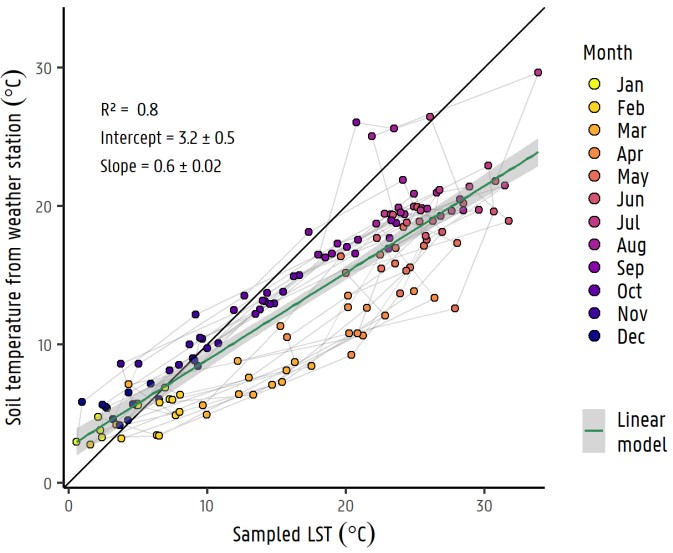

**Figure 4 In situ measurements at weather stations vs sampled LST values. The linear model that was used to correct raster LST values is included.**






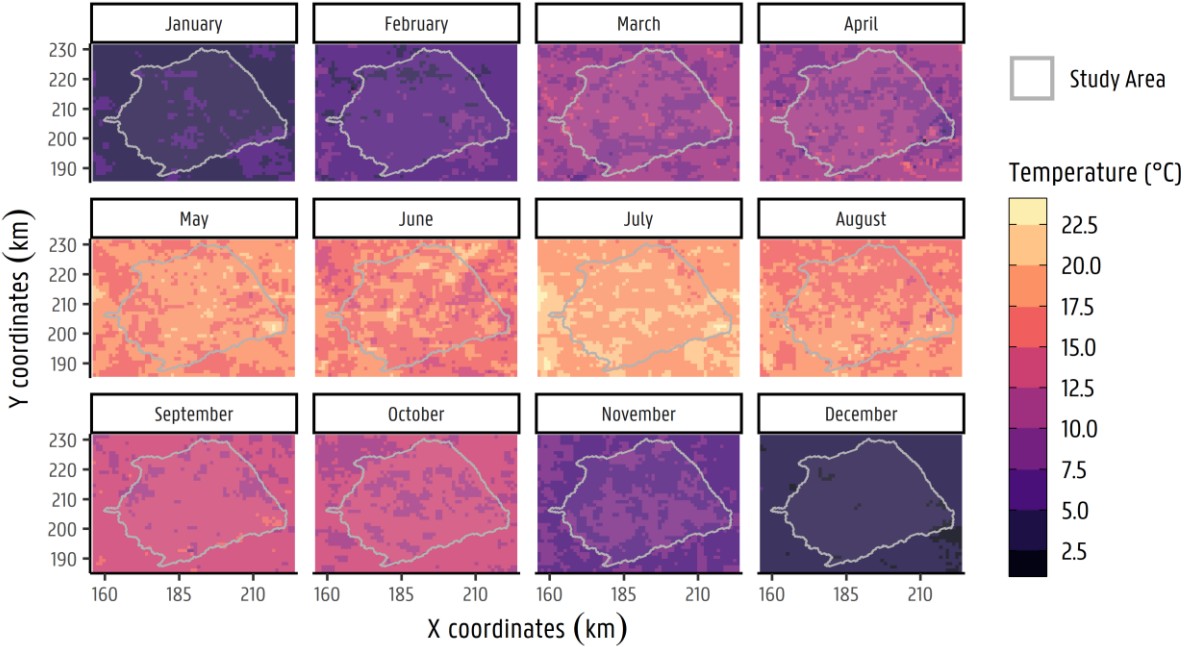

**Figure 5 Corrected MODIS monthly average LST with a 1 km resolution for year 2001 within the Neogene aquifer model (NAM) boundary.**





**Figure 6 Temperature-time curve for each author scaled to the 1961-1990 Belgian average for the past 10,519 years.**





**Figure 7 Temperature-time curve to use as input in transient heat-transport simulation indicating time step sizes for the last 10519 years.**





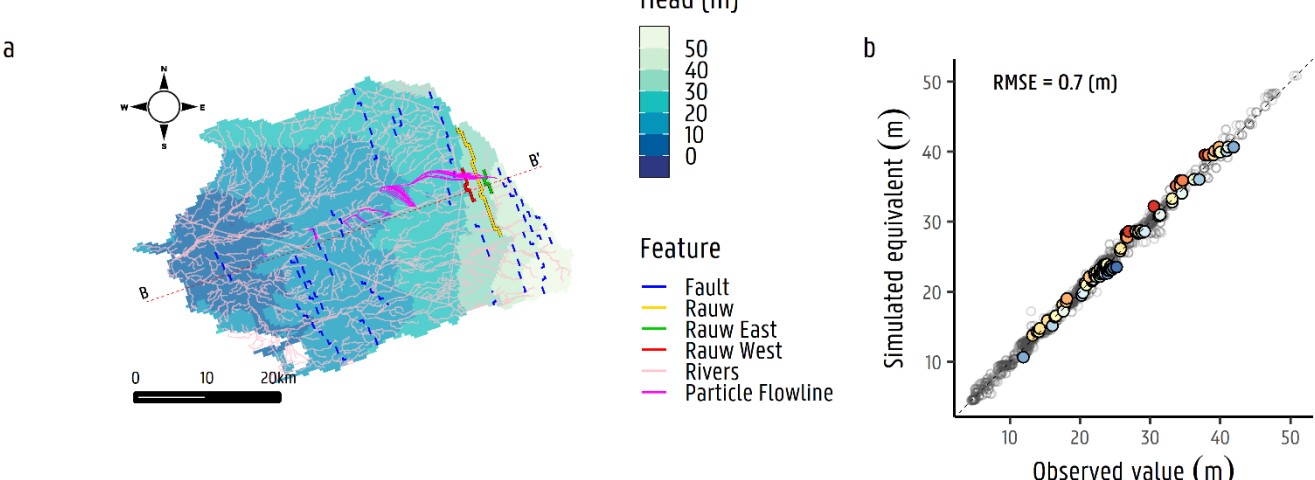

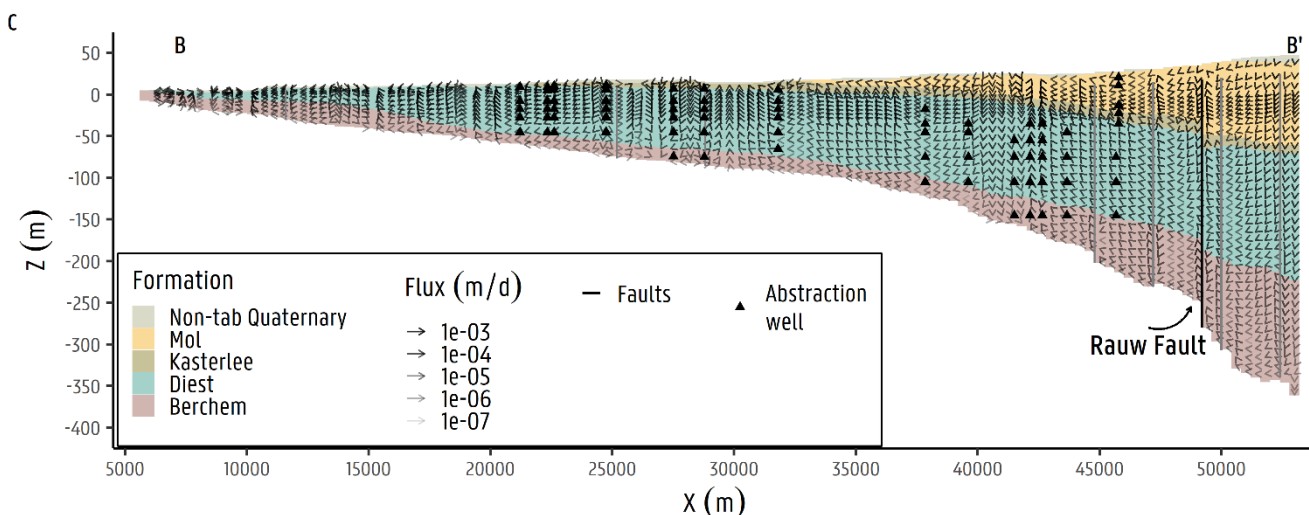

**Figure 8 Groundwater flow model results adapted from Casillas-Trasvina et al., (accepted for publication, 2021): a) The hydraulic head distribution over the Nete catchment. b) Scatter plot of simulated equivalents and observed hydraulic heads. c) Cross-section B-B' with arrows indicating the flow magnitude (m/d) and direction.**





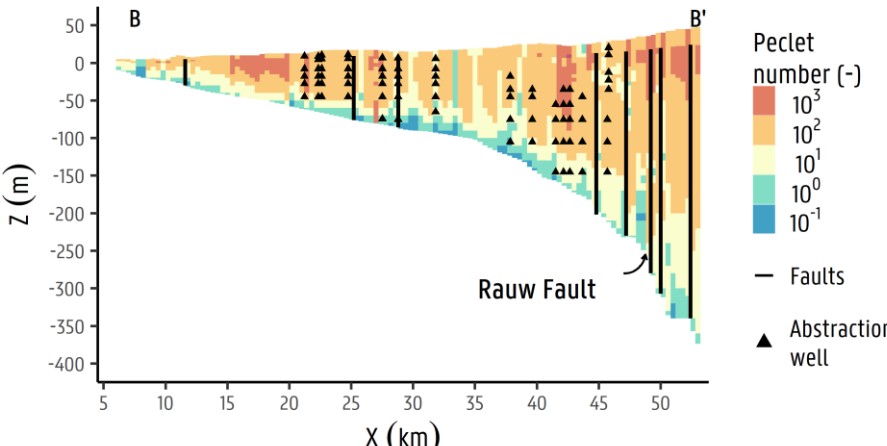

**Figure 9 Peclet numbers (Pe) derived from groundwater model results presented by Casillas-Trasvina et al., (accepted for**
**publication, 2021). Note the considerable lower values (Pe ≤ 1) in the deepest parts of the Neogene aquifer corresponding to the**
**Berchem & Voort Sands.**





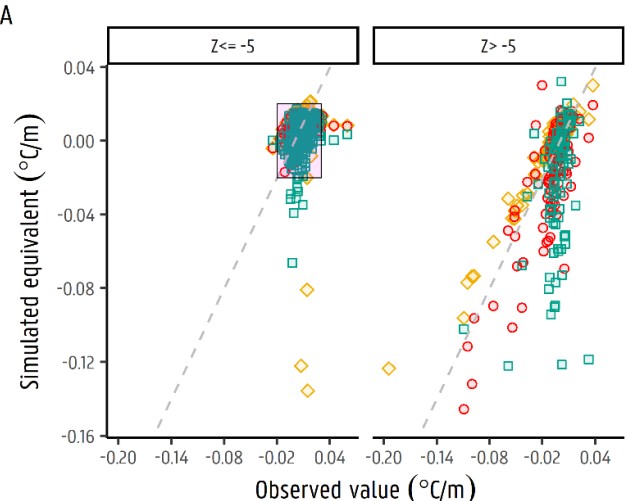

| Model | RMSE Flow (m) | RMSE Temp (°C/m) |
|---|---|---|
| Paleo RPS | 0.70 | 0.03 |
| Montly LST | 0.84 | 0.04 |
| Steady State | 0.84 | 0.08 |

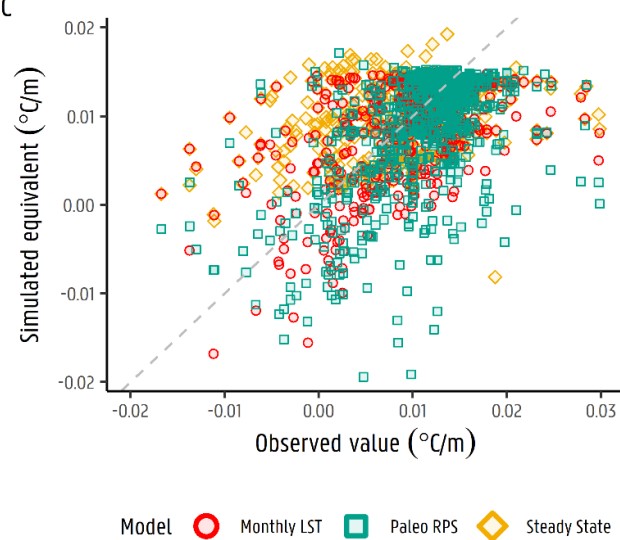

**Figure 10 Simulated vs observed temperature gradients showing results from the steady-state model, the monthly LST model and reconstructed paleo-temperatures included as input into the transient model. a) Distinction made between observations below and above z = -5 m a.s.l.; b) RMSE performance of different models in terms of hydraulic head (flow model, m), and temperature gradient (°C/m). c) A zoom into the simulated and observed temperature gradients shown in Figure a.**






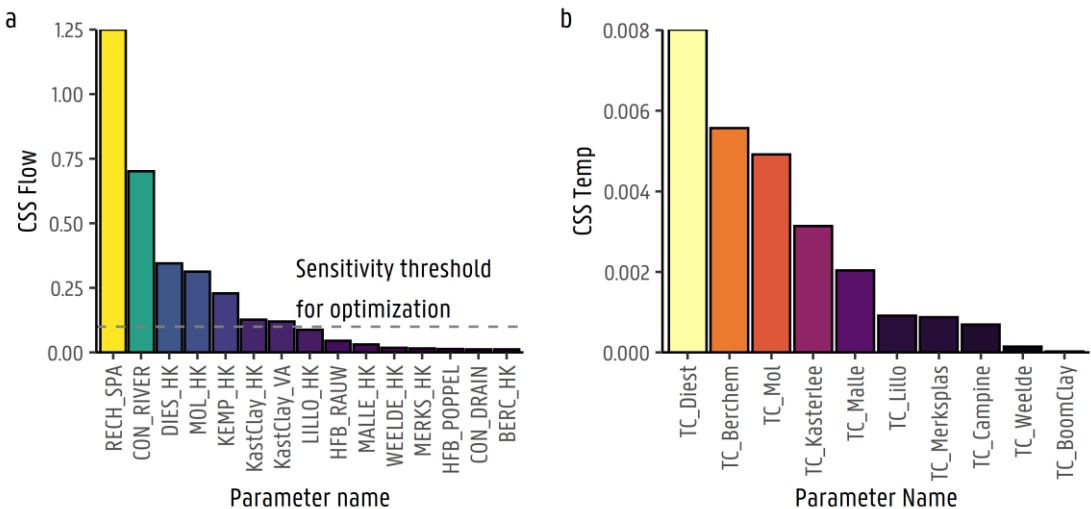

**Figure 11 Composite scaled sensitivities for: a) the hydraulic head observations and, b) the temperature gradients derived from TD observations.**


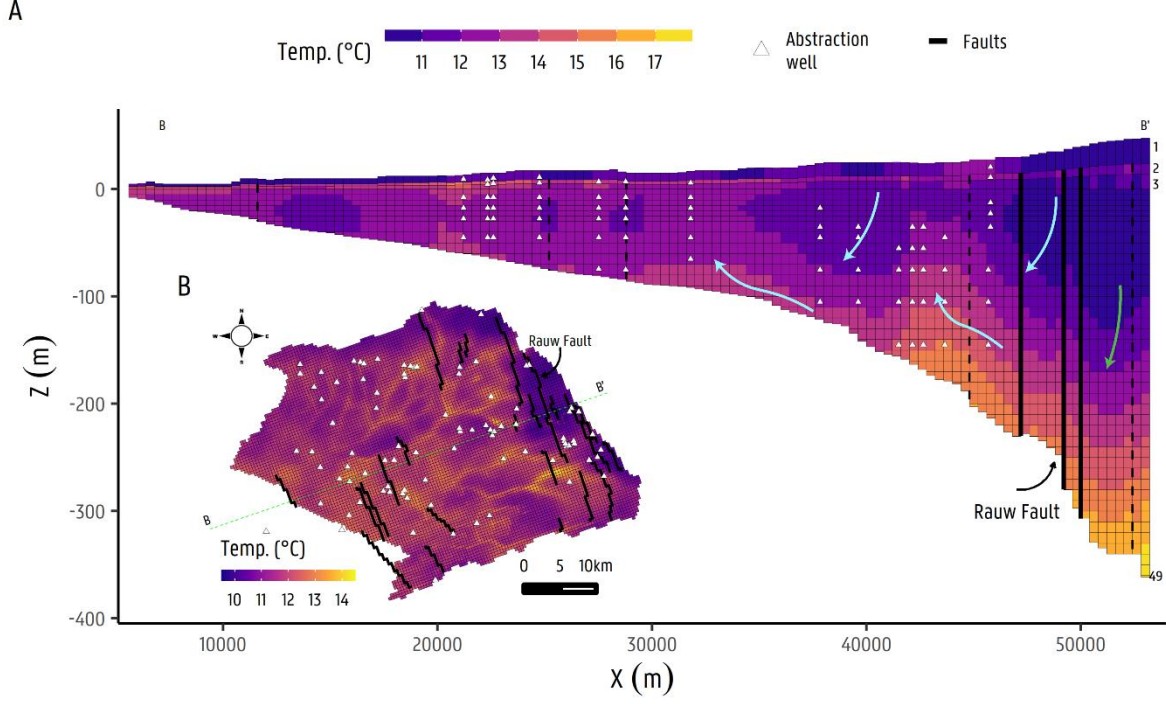

**Figure 12 Temperature distribution at the last transport step for the paleo-temperature transport model in a) Cross section. Green and cyan arrows indicate flow systems and fluxes in the east and west side of the Rauw Fault; and b) plan view from model layer 10, indicating cross section (B-B') for figure a.**









**Figure 13 a) Map showing selected TD profile locations along the approximate flow direction (magenta circles): R-13d, R-54c, R-11c, R15f, R-2c, R34c, R-1b, R51c and R-43c (background map: © OpenStreetMap contributors, 2020. Distributed under the Open Data Commons Open Database License (ODbL) v1.0b) Temperature-depth profiles of selected wells showing simulated values for the paleo-temperature model (baseline), and for simulation cases 2 and 3 (only thermal conduction and without faults, respectively). Modelled TD profiles for every time-step of the baseline case are also included (in gray, paleo modelled).**








**Figure 14 Cross sections and plan views of model layer 10 showing the differences between the baseline model minus both cases: a & b) for the thermal conduction only case, and c & d) for the temperature distribution without faults.**





**Appendices**

**Figure A1 – Measured and simulated temperature-depth profiles for all locations across the Neogene aquifer, within the Nete Catchment.**
