# Peer review of "Characterizing groundwater heat-transport in a complex lowland aquifer using paleo-temperature reconstruction, satellite data, temperature-depth profiles, and numerical models."

_Hydrology and Earth System Sciences, 2021_

## Referee Comment (RC1)

Review of the manuscript

*"Characterizing groundwater heat-transport in a complex lowland aquifer using paleo-temperature reconstruction, satellite data, temperature-depth profiles, and numerical models"*

Manuscript ID: hess-2021-586.

**General comments:**

The manuscript introduces a numerical groundwater flow and heat transport model for the region of northeast Flanders, Belgium. It is a very detailed case study that is very useful for other hydrogeologist studying this region (and potentially other students working on setting up comparable models for other regions), but it is sometimes hard to follow for someone not familiar with the site. As I am not an expert in numerical modeling I cannot speak to this aspect of the study, however at this stage there are some shortcomings in the structure of the paper and in the selection of surface temperature data as an upper boundary condition.

I am particularly worried about the lack of separation between satellite derived land surface temperature, air temperature, and (ground) surface temperature/soil temperature (which would be relevant for the subsurface thermal regime). While these temperatures correlate, they are not the same.

I am furthermore not clear of the benefit of running the model based on temperature time series starting in 8500 BC – we do not have accurate data for this far back and more importantly, the model reaches down only 400 m at most, at this depth surface temperatures from maybe a couple hundred years ago are dominating the thermal regime. Hence, I must recommend focusing rather on high-frequency and more accurate data that's a bit more recent than on long time-series such as paleo-temperatures (e.g., climate reanalysis data such as ERA 5 or MERRA-2 or GISS Model E2)

On a smaller note – I had a hard time really understanding study area. The authors could consider renaming their boreholes and faults to somethings more intuitive. Right now, the boreholes are called R-51 which doesn't give a ton of info. I'd suggest something like borehole1_2019 s in first borehole from the left, measured in 2019. They also often refer to parameter names or other sites in the text that are not mentioned anywhere else and thus pretty irrelevant and non-gettable for a reader not already familiar with either the study site or the model (e.g. Fig 11 or Line 401).

**Detailed comments:**

1. Line 55 "However, in many cases […], when there are multiple unknowns on flux exchanges and thermal and hydraulic properties, it is likely to reproduce temperature correctly …" What is 'it'?
2. Line 65: "Several studies where temperature profiles are measured are typically used for qualitative interpretations of the effects of anthropogenic stressors." Do you mean "… are measured, they are typically used…"?
3. Line 110 – discussion of the ON-MOL-2 site seems highly irrelevant. It is not shown in any of the figures or discussed in the results and discussion sections.
4. Chapter 3.1.1.
   - reference to Figure A1.
   - I'd also appreciate a clearer indication of the year (and maybe even month for seasonality impacts) the measurements were taken in Fig A1 and Fig 2
   - Give some info here about the depth of these TD-profiles
   - Where any of the measurements repeated or are the 35 TD profiles taken in 35 different locations?
5. Chapter 3.1.2. what is the depth of the soil temperature measurements?
6. Chapter 3.2.3.
   - While not really relevant to the study the suggested missions (Landsat, ASTER, MODIS and Copernicus) and the discussion of those are worded in a way that could be misleading. Specifically, Landsat LST are available globally, just not daily, and with Copernicus I guess you are referring to the second gen Meteosat?
   - Please give info what kind of MODIS data you are using. Day or night? Terra or Aqua (i.e., MOD or MYD)?
   - Line 169 "From this sampling, it is clear that the raster LST images overestimate the peak temperature values, mostly during the summer months" you are comparing LST to air and soil temperature. They are not overestimating air and soil temperature; they simply measure a different (but correlated) value! Hence you are not correcting LST, you are estimating (soil? air? I'm not clear) temperatures from LST.
   - Line 175 LST is fitted to soil or air temperature?
7. Chapter 3.1.4 – are these temperatures air, soil or LST? I assume air but should be made clear. Also, as mentioned in the main comment I'm doubtful about the need for such a long (and inaccurate) time series. I'd suggest keeping the 90 time steps but for a shorter time series.
8. Line 212 – what depth is the bottom boundary? It is shown in the figure, but I'd appreciate a short sentence here
9. Line 228 – bracket is never closed (')' missing).
10. Line 245 "(Databank Ondergrond Vlaanderen; DOV).. The model" one dot too much.

11. Line 317 – how where LST downscaled?
12. Chapter 4 – generally I'm not sure for what date your model gives output. Dec 2019? How are you then comparing this to the TD-profiles from previous years?
13. Line 356 – why are you comparing temperature gradients instead of absolute temperatures when comparing these models? And how did you (statistically) calculate the gradients?
14. Line 360 – Why did you split the model at -5m.a.s.l? In temperature modeling meters below surface are of course much more relevant hence I believe a split at maybe 20 m below ground would be more useful, above those seasonal impacts may be way more relevant than anything else.
15. Line 387 – Space missing between "Figure 12b.Figure 12a"
16. Line 401 – I'm not really seeing any blue dashed lines in Fig 12 a (or b) – also seems highly irrelevant as "HFB_094" etc are never mentioned again.
17. Line 426 – are measured and observed shown for the same time?
18. Line 461 – how does this strongly support the statement that heat transport is driven by advection? You are comparing two models, you have not shown me yet which one is more accurate in these locations, have you?
19. Chapter 5
    o Could you give some sort of accuracy assessment of your models? This info is in part scattered throughout the manuscript, but as a reader I would benefit from an extra chapter just giving me all the info in one location which models are how accurate in which locations (near surface, near faults, etc.)
    o I'd like to see a (qualitative) cost-benefit analysis of your model. Is the extra time and computing power needed to run the long time series worth the effort or is an analytical solution basically as good and much faster, less costly in computing power.
20. Fig 1 – really hard to read. It was rather low res in my pdf and the colors. I cannot read any of the writing on the map where LST is shown. The legend is missing the yellow marks showing boreholes, are temperatures shown as discrete values (as shown on the legend) or as a stretched values along a color ramp?
21. Some of the methodology figures could be moved to the Appendix (Fig 1-9)
22. Fig 8 – is meters here m.a.s.l. or meters below ground?
23. Fig 10a – make it square so it's easier to cmompare. Unit for z<= -5 and z>-5 missing. Also use ≥ instead of >=
24. Fig 11 – the parameter names along the x-axis mean nothing to me, they are not mentioned in the manuscript anywhere else.
25. Fig 12 – the cyan and green arrows are barely distinguishable in my pdf
26. Fig 13 – pink circles not in legend, include B-B' line, in b the names of the boreholes are different than anywhere else (i.e. the small case letters at the end)

27. Fig 14 – give the names of the wells so I can compare.
28.

---

## Author Comment (AC1)

Response to RC-1 review of the manuscript:
*"Characterizing groundwater heat-transport in a complex lowland aquifer using paleo-temperature reconstruction, satellite data, temperature-depth profiles, and numerical models"*
Manuscript ID: hess-2021-586.

**General comments:**

The manuscript introduces a numerical groundwater flow and heat transport model for the region of northeast Flanders, Belgium. It is a very detailed case study that is very useful for other hydrogeologist studying this region (and potentially other students working on setting up comparable models for other regions), but it is sometimes hard to follow for someone not familiar with the site. As I am not an expert in numerical modeling I cannot speak to this aspect of the study, however at this stage there are some shortcomings in the structure of the paper and in the selection of surface temperature data as an upper boundary condition.

1. I am particularly worried about the lack of separation between satellite derived land surface temperature, air temperature, and (ground) surface temperature/soil temperature (which would be relevant for the subsurface thermal regime). While these temperatures correlate, they are not the same.

Response: Indeed, these temperatures are not the same. The satellite derived land surface temperature (LST) was used to determine a spatially distributed temperature over the whole study area. The soil temperature was then used to estimate the LST spatially distributed temperatures using a linear regression model.

Mentioning the air temperature in the text might result confusing. In the revised version, section 3.1.2 and 3.1.3 will clarify these aspects in more detail.

2. I am furthermore not clear of the benefit of running the model based on temperature time series starting in 8500 BC – we do not have accurate data for this far back and more importantly, the model reaches down only 400 m at most, at this depth surface temperatures from maybe a couple hundred years ago are dominating the thermal regime. Hence, I must recommend focusing rather on high-frequency and more accurate data that's a bit more recent than on long time-series such as paleo-temperatures (e.g., climate reanalysis data such as ERA 5 or MERRA-2 or GISS Model E2).

Response: Certainly, no accurate data exists for this far back. However, temperature reconstructions from an exhaustive list of published works (i.e. Buntgen et al., 2011; Casty et al., 2007; D'Arrigo et al., 2005, 2006; Esper et al., 2014; Glaser and Riemann, 2009; Ljungqvist et al.,

2019; Luterbacher et al., 2016; Mann, 2002; Mann et al., 2009; Marsicek et al., 2018; Tingley and Huybers, 2013) that are widely accepted have been used to infer a temperature value to be implemented as initial condition for the heat-transport model top boundary condition.

To increase the accuracy of the simulated temperature, hence a good simulated vs observed temperature performance, the simulated temperature has to be able to stabilize (or reach a steady-state) and for this we require to estimate these initial conditions (initial temperature values for each stress period).

To demonstrate the importance of performing the paleo-temperature simulations to provide initial conditions for each stress period, a test was performed. Simulations were performed where the initial top boundary condition was disturbed by increasing 1 degree at various time steps (years before present i.e. 10000yr, 9000yr, 8000yr, 7000yr, 6000yr, 5000yr, 4000yr, 3000yr, 2000yr, 1000yr, 900yr, 800yr, 700yr, 600yr, 500yr, 400yr, 300yr, 200yr, 100yr) and for the remaining time-steps of the simulation period. The simulated temperature-depth (TD) profile from all these models was obtained for the same location (on well R-54f). The results of this test are shown in Figure 1. In Figure 1a, the simulated TD profile at the last time step (present time) from a normal (undisturbed) forward model run is shown. In Figure 1b, the differences between the disturbed and normal (undisturbed) simulated TD profiles are shown, pointing out the time required for the model to reach a steady-state given a change of 1 degree in the top boundary temperature condition. This shows that a couple of hundreds of years are still not sufficient to achieve a steady-state condition. For the purposes of our study, it is important to be able to reach this stability at these depths of the aquifer.

This supports the use of a relatively long time series which is actually a process that is not computationally intensive. The model in total has 337 stress periods (accounting the paleo-temperature time series for 111 steps, and 226 for the LST stress periods) and runs in around 40 minutes. For the 9 time steps in the stress period between 10000yr and 2000yr before present (in steps of 1000 years) it requires around a minute (approximately 64 seconds) to compute, offering the considerable temperature gain is of up to 30% per temperature degree of change at the temperature top boundary condition. Having a higher frequency temperature distribution is not within the scope of this work, as we are more interested in knowing what improvement the implementation of an additional state variable may bring to the conceptualization of the groundwater system rather than aiming for higher frequency results, for which monthly time-steps seem reasonable.

**Commented [BK1]:** I would reverse this list, start with 10000, because this is also the sense/direction of the simulation

**Commented [BK2]:** Try to be consistent in time period designation

**Commented [BK3]:** This sentence reads a bit weird, try to rephrase

[Figure]

*Figure 1 a) Temperature depth (TD) profile simulated at observation well (i.e. R-54f) at the end of the coupled groundwater flow and heat transport undisturbed simulation (full time period up till the present time). b) Temperature difference at the same observation well between the TD profile at the end of each disturbed simulation (from the indicated time up till the present time) minus the undisturbed simulation.*

3. On a smaller note – I had a hard time really understanding study area. The authors could consider renaming their boreholes and faults to somethings more intuitive. Right now, the boreholes are called R-51 which doesn't give a ton of info. I'd suggest something like borehole1_2019 s in first borehole from the left, measured in 2019. They also often refer to parameter names or other sites in the text that are not mentioned anywhere else and thus pretty irrelevant and non-gettable for a reader not already familiar with either the study site or the model (e.g. Fig 11 or Line 401).

Response: In the borehole naming, the year of measurement is something that can be included (e.g. R-51x_2018). However, the name coding (for boreholes and faults ,e.g. R-51x, HFB_XX, and the naming of the parameters) should stay as this is the coding for the regional boreholes in the study area. In other works in the region this naming is similarly used as it works as a reference as these are all formal ID's and accepted terminology, for boreholes that can even be looked up in the Databank Ondergrond Vlaanderen (DOV) Verkenner. For this reason, the namings should remain as they are..

**Detailed comments:**

1. Line 55 "However, in many cases […], when there are multiple unknowns on flux exchanges and thermal and hydraulic properties, it is likely to reproduce temperature correctly …" What is 'it'?

Response: "It" indicates 'calibration' but was not included in the text. The sentence is rephrased to:

"However, as demonstrated by Bravo et al. (2002), Kurtz et al. (2014), Irvine et al. (2015) and Delsman et al. (2016), when there are multiple unknowns on flux exchanges and material (i.e. thermal and hydraulic) properties, the calibration using hydraulic head and temperature observation as targets is likely to reproduce temperature correctly in spite of a potential incorrect representation of fluxes (Schilling et al., 2019)."

2. Line 65: "Several studies where temperature profiles are measured are typically used for qualitative interpretations of the effects of anthropogenic stressors." Do you mean "… are measured, they are typically used…"?

Response: Agree, the text should read as pointed out. Changes made accordingly.

"Several studies where temperature profiles are measured, they are typically used for qualitative interpretations […]"

3. Line 110 – discussion of the ON-MOL-2 site seems highly irrelevant. It is not shown in any of the figures or discussed in the results and discussion sections.

Response: Agree. The reference and discussion of the ON-MOL-2 site is not relevant here. It is removed from this section and in further parts of the manuscript as it is really not important to point out.

4. Chapter 3.1.1.
   1) Reference to Figure A1.

Response: Agree, reference to Figure A1 is included in line 150.

2) I'd also appreciate a clearer indication of the year (and maybe even month for seasonality impacts) the measurements were taken in Fig A1 and Fig 2.

Response: Agree, the month of the when the measurement was taken will be included in Table 1.. The date in figure 2 is also included for the groups of wells.

3) Give some info here about the depth of these TD-profiles

Response: In line 149 the range of depths to these TD profiles is included. Reference to Figure A1 is included in line 150, to show the depth to each of these filters.

4) Were any of the measurements repeated or are the 35 TD profiles taken in 35 different locations?

Response: A total of 23 different locations were measured with a total of 35 filters (some filter (i.e. R-12 e-f, R-15 b-f, R-16 c-d, R-1 b-c, R-54 a-c-f) are in the same borehole). Clarification of this is included in line 150.

5. Chapter 3.1.2. what is the depth of the soil temperature measurements?

Response: The depth of the soil temperature measurements is 5 of cm. This in included in the revised version of the manuscript.

6. Chapter 3.2.3.
   1) While not really relevant to the study the suggested missions (Landsat, ASTER, MODIS and Copernicus) and the discussion of those are worded in a way that could be misleading. Specifically, Landsat LST are available globally, just not daily, and with Copernicus I guess you are referring to the second gen Meteosat?

Response: Agree. The discussion in lines 163-166 is rephrased to avoid misleading the reader. And yes, for the Copernicus, we refer to the second gen Meteosat, which is included in the edited version of this section.

   2) Please give info what kind of MODIS data you are using. Day or night? Terra or Aqua (i.e., MOD or MYD)?

Response: Agree. Info is included in line 167 about the MODIS data used (terra, day and night averages).

   3) Line 169 "From this sampling, it is clear that the raster LST images overestimate the peak temperature values, mostly during the summer months" you are comparing LST to air and soil temperature. They are not overestimating air and soil temperature; they simply

measure a different (but correlated) value! Hence you are not correcting LST, you are estimating (soil? air? I'm not clear) temperatures from LST.

Response: Agree. The section from line 169 to 175 is rephrased, as it is not corrected but a soil temperature is estimated from the LST. This is corrected across the manuscript where 'overestimation' is used. Soil temperatures are those estimated, and it is also included here, and across the manuscript where missing.

4) Line 175 LST is fitted to soil or air temperature?

Response: Fitted to soil temperature, included here in line 175 for clarification.

7. Chapter 3.1.4 – are these temperatures air, soil or LST? I assume air but should be made clear. Also, as mentioned in the main comment I'm doubtful about the need for such a long (and inaccurate) time series. I'd suggest keeping the 90 time steps but for a shorter time series.

Response: These are LST values. It is clarified in the text, in line 183. The rest of this comment has been already addressed the *General comments section – comment 2*.

8. Line 212 – what depth is the bottom boundary? It is shown in the figure, but I'd appreciate a short sentence here.

Response: Agree. Included in line 212 the depth to the bottom of the aquifer/top of Boom Clay, from near the surface down to around 400 meters deep.

9. Line 228 – bracket is never closed (')' missing).

Response: Agree. Included closing parenthesis ")" in "[…] above the 'Kempen' aquifer system), Weelde […]".

10. Line 245 "(Databank Ondergrond Vlaanderen; DOV).. The model" one dot too much.

Response: Agree. Deleted repeated dot.

11. Line 317 – how where LST downscaled?

Response: The 1-km LST rasters were downscaled to 100x100 using the {raster} package, function disaggregate in R. Then, the mean value of the downscaled temperatures within a modelling cell was estimated and set in the MODFLOW model. This explanation is to be included in line 319.

12. Chapter 4 – generally I'm not sure for what date your model gives output. Dec 2019? How are you then comparing this to the TD-profiles from previous years?

Response: The model returns a stack of 3D matrixes, one for each time-step in every stress period. Hence, every measured TD profile is compared with its respective simulated temperature based on its spatial and temporal (i.e. month and year) location.

13. Line 356 – why are you comparing temperature gradients instead of absolute temperatures when comparing these models? And how did you (statistically) calculate the gradients?

Response: Temperature/geothermal gradients are more important than absolute temperature values as they provide more insights in groundwater fluxes. The gradient is calculated by fitting a cubic smoothing spline with a knot distance of 1 meter.

14. Line 360 – Why did you split the model at -5m.a.s.l? In temperature modeling meters below surface are of course much more relevant hence I believe a split at maybe 20 m below ground would be more useful, above those seasonal impacts may be way more relevant than anything else.

Response: Agree, this is in fact the reason why the split is done, as mentioned in lines 361-366. In the first model layer the temperature top-boundary condition is imposed, so those values above 0 are basically sampling this boundary condition. Those below this depth are those temperatures actually simulated by the model and hence supports the split at this depth.

15. Line 387 – Space missing between "Figure 12b.Figure 12a".

Response: Agree. Spacing is added in the revised version where is has been pointed out.

16. Line 401 – I'm not really seeing any blue dashed lines in Fig 12 a (or b) – also seems highly irrelevant as "HFB_094" etc are never mentioned again.

Response: Agree. The figure shows 'black dashed lines' and in the text is mentioned 'blue dashed lines'. This error is corrected in the text. Mentioning the faults (HFB_094 and so on) seems indeed, not really necessary and is to be removed in the revised version.

17. Line 426 – are measured and observed shown for the same time?

Response: No. As explained above in point 12 (comment on Chapter 4), each of the measured TD profiles are compared with  respective simulated TD profiles based on the month and year, and of course, location.

18. Line 461 – how does this strongly support the statement that heat transport is driven by advection? You are comparing two models, you have not shown me yet which one is more accurate in these locations, have you?

Response: Chapter *4.3.1 Thermal conduction* case represents heat being only transported across the aquifer via thermal conduction, as the advective mechanism is removed. The intention of this case is not to determine which case is more or less accurate than the other, as it is known that advective transport exists and not including it would not be realistic, but to highlight the spatially importance or dominance of the advective transport mechanism across the aquifer. Several locations are already mentioned in the discussion and clearly shown in Figure 14, where the differences point out that the *thermal conduction* case is less accurate.

19. Chapter 5
    1) Could you give some sort of accuracy assessment of your models? This info is in part scattered throughout the manuscript, but as a reader I would benefit from an extra chapter just giving me all the info in one location which models are how accurate in which locations (near surface, near faults, etc.).

Response: Agree. A further section will be added in 4.3.3 to briefly describe the models, their accuracies in specific location.

    2) I'd like to see a (qualitative) cost-benefit analysis of your model. Is the extra time and computing power needed to run the long time series worth the effort or is an analytical solution basically as good and much faster, less costly in computing power.

Response: As previously in the *General comments section - point 3* of this response list details, this comments is already addressed, see below. Additionally, the benefit of using the paleo-temperature input will be further mentioned in the conclusions section.

The use of a relatively 'long' time series is actually a process that is not computationally intensive as the time-steps in the stress periods before the year 500 are quite spaced. The model in total has 337 stress periods (accounting the paleo-temperature time series for 111 steps, and 226 for the LST stress periods) and runs in around 40 minutes. For the 9 time steps in the stress period between 10000yr and 2000yr before present (in steps of 1000 years) it requires around a minute (approximately 64 seconds) to compute, offering the considerable temperature stabilization of up to 30% per temperature degree of change in the temperature top boundary condition.

The use of a fully-3D numerical model for flow and transport in groundwater is widely supported for this type of problem as is it relatively easy to implement and their ability to solve the partial differential equations governing these phenomena where an analytical solution would perhaps

**Commented [BK4]:** Try to harmonize indication of timing: in full, or Kyr or kyr…

**Commented [BK5]:** Sentence sounds a bit weird, try to rephrase

present complications for accounting for the spatio-temporal stresses (e.g. vertical/horizontal in/out fluxes across time and space).

20. Fig 1 – really hard to read. It was rather low res in my pdf and the colors. I cannot read any of the writing on the map where LST is shown. The legend is missing the yellow marks showing boreholes, are temperatures shown as discrete values (as shown on the legend) or as a stretched values along a color ramp?

Response: The resolution of the file seems to have been reduced significantly in the pdf version. The actual version has actually a quite large resolution. Nevertheless, the font sizes in the map will be increased and made clearer. The legend will now include the location of the boreholes. And yes, the temperature is shown as stretched values along the color ramp. The color ramp in the legend will be modified to clarify this.

21. Some of the methodology figures could be moved to the Appendix (Fig 1-9).

Response: Agree. For brevity in the final version Figures 3, 4, 5, 6, 9 and 11 are moved to the appendix in the revised version.

22. Fig 8 – is meters here m.a.s.l. or meters below ground?

Response: Meters here are 'm.a.s.l.'. This is changed in the figure labels when required in the revised version of the manuscript.

23. Fig 10a – make it square so it's easier to cmompare. Unit for z<= -5 and z>-5 missing. Also use ≥ instead of >=

Response: Agree, Figure 10a will be modified to fit a squared shape. The units in the legend of Figure 10a are also changed accordingly.

24. Fig 11 – the parameter names along the x-axis mean nothing to me, they are not mentioned in the manuscript anywhere else.

Response: Agree. A table is included where Figure 11 is mentioned to reference and clarify the names of the parameters.

25. Fig 12 – the cyan and green arrows are barely distinguishable in my pdf

Response: Agree. Perhaps it is related to the quality of the pdf file. Still, the arrows will be increased in size to be more noticeable.

26. Fig 13 – pink circles not in legend, include B-B' line, in b the names of the boreholes are different than anywhere else (i.e. the small case letters at the end).

Response: Agree. Although the pink circles are included in the caption of the figure, these will be included in the legend of the figure. A B-B' line is included. The names of the boreholes are correct, as these indicate the actual filter in which the TD profile is measured. Clarification on the naming of the boreholes and their filters have been included in section 3.1.1 and is thoroughly clarified across the manuscript.

**Commented [BK6]:** Check my previous comment on this.

27. Fig 14 – give the names of the wells so I can compare.

Response: Those shown in this figure are abstraction wells, and not observation wells used to measure TD profiles. The location of the TD profile boreholes, names and filters is included for those wells in and nearby this cross section for reference.

---

## Author Comment (AC2)

**Response to RC-2 review of the manuscript:**

*"Characterizing groundwater heat-transport in a complex lowland aquifer using paleo-temperature reconstruction, satellite data, temperature-depth profiles, and numerical models"*
Manuscript ID: hess-2021-586.

General comments:

The paper by Casillas-Trasvina et al. aims to simulate the heat transport in the Neogene aquifer. Although this manuscript collects a lot of data, which can be useful for other works, it is difficult to read and the presentation should be more concise and to the point. I have the following main comments:

1. It is not clear why the model needs paleo-temperature to work. Also, I really doubt that simulating 10519 years yields meaningful results, given that the flow model is stationary.

Response: To increase the accuracy of the simulated temperature and hence a good simulated vs observed temperature performance, the simulated temperature has to be able to stabilize (or reach a steady-state) and for this we require to estimate initial conditions (initial temperature values for each stress period).

To show the importance of performing the paleo-temperature simulations to provide initial conditions for further stress periods, a test was performed. (Disturbed) simulations increasing 1 degree at the initial top boundary condition at various time steps (years before present i.e. 10000yr, 9000yr, 8000yr, 7000yr, 6000yr, 5000yr, 4000yr, 3000yr, 2000yr, 1000yr, 900yr, 800yr, 700yr, 600yr, 500yr, 400yr, 300yr, 200yr, 100yr) and for the remaining of the simulation period were performed. The simulated temperature-depth (TD) profile from all these models was obtained for the same location (on well R-54f). The results of this test are shown in Figure 1. On Figure 1a, it is shown the simulated TD profile at the last time step (present time) from a normal (undisturbed) forward model run. On Figure 1b, the figure shows the differences between the disturbed and normal (undisturbed) simulated TD profiles, pointing out to the time required for the model to reach a steady-state given a change of 1 degree in the top boundary temperature condition. This supports the use of a relatively long time series which it is actually a process that is not computationally intensive. The model in total has 337 stress periods and runs in around 40 minutes For the 9 time steps in the stress period between 10000yr and 2000yr before present (in steps of 1000 years) it requires around a minute (approximately 64 seconds) to compute offering a temperature stabilization of up to 30% per temperature degree of change in the temperature top boundary condition.

Similarly as done in previous works in the area (Casillas-Trasvina et al., 2021; Gedeon, 2008; Rogiers et al., 2015), and as indicated in the body if the manuscript (section 3.2.1 Conceptual model, line 215), the aquifer is assumed to be in dynamic equilibrium, with no long-term trends in groundwater fluxes, which allows us to simulate the groundwater flow in steady-state. For the purposes of our research we find this assumption acceptable.

[Figure]

*Figure 1 a) Temperature depth (TD) profile simulated at observation well (i.e. R-54f) at the end of the coupled groundwater flow and heat transport undisturbed simulation (present time). b) Temperature difference at the same observation well between the TD profile at the end of each disturbed simulation (present time) minus the undisturbed simulation.*

Casillas-Trasvina, A. C., Rogiers, B., Beerten, K., Wouters, L. and Walraevens, K.: Exploring the hydrological effects of normal faults at the boundary of the Roer Valley Graben in Belgium using a catchment-scale groundwater flow model, Hydrogeol. J., (0123456789), doi:10.1007/s10040-021-02423-y, 2021.

Gedeon, M.: Neogene Aquifer Model., SCK-CEN external report ER-48, 100 pages. Report prepared by SCK•CEN in the framework of ONDRAF/NIRAS programme on geological disposal, under contract CCHO- 2004-2470/00/00, DS 251-A51.2008.

Rogiers, B., Labat, S. and Gedeon, M.: An assessment of dilution tests and ambient temperature logging for quantifying groundwater flow in the Neogene aquifer., SCK-CEN Extern. Rep., 2015.

2. Peclet number: I have not understood why the Pe number is smaller than 1 in some parts of the aquifer. I really doubt that transport can be diffusive in aquifers.

Response: Peclet numbers in the large majority of the aquifer are well above a value of 1 (Pe >> 1). However, in the Berchem & Voort Formation, Peclet numbers are above and around 1 in the majority of the formation and increasing up to around 10 towards the Diest Formation, right above it. The values of Pe ≤ 1 are very few, mostly near to 1, and found near the bottom of the aquifer where groundwater flow velocities are at the lowest as in these areas the fluxes occur near the bottom no-flow boundary of the model. Research has been previously performed as summarized by Vandersteen et al. (2014), pointing out the low hydraulic conductivity values for the Berchem & Voort Formation (as low as 0.02 m/d), and thus not particularly acting as a barrier/clay, but with modest advective/diffusive behavior.

Vandersteen, K., Gedeon, M. and Beerten, K.: A synthesis of hydraulic conductivity measurements of the subsurface in Northeastern Belgium, Geol. Belgica, 17(3–4), 196–210, 2014.

Minor comments:

1. Ln. 85: a full stop is missing before This work.

Response: Included a full stop "." after "techniques" in "…techniques. This work…"

2. Figure 2: Please write also in the caption the source of the map

Response: Reference to the map is included in the caption, reading as follows:

> '*Figure 2. Plan view of the study area as discretized in the second layer of the numerical model. It indicates faults (emphasis on the highlighted Rauw Fault), cross section, temperature-depth profile locations, and modelled formations derived from the hydrogeological 3D model from Deckers et al., (2019).* '

Deckers, J., De Koninck, R., Bos, S., Broothaers, M., Dirix, K., Hambsch, L., Lagrou, D., Lanckacker, T., Matthijs, J., Rombaut, B., Van Baelen, K. and Van Haren, T.: Geologisch (G3Dv3) en hydrogeologisch (H3D) 3D-lagenmodel van Vlaanderen. Studie uitgevoerd in opdracht van: Vlaams Planbureau voor Omgeving (Departement Omgeving) en Vlaamse Milieumaatschappij.,., 2019.

3. Ln. 137: avoid abbreviation. Use was not instead of was't.

Response: "Wasn't" changed to "was not".

4. Figure 3: the figure is not clear. It should be somehow indicated that b) is the inset in a). Also the curves are not clear and the data from Eindhowen look completely missing.

Response: Agree. Indication 'b' in the red square of figure a is included. In the caption, it is indicated that figure b is an inset of figure a. Additionally, the curves are replotted to be more clearly shown (thicker lines) and showing the data from the station Eindhoven.

5. Ln. 162: what is i.a. ?

Response: i.a. (inter alia) is Latin for "among other things".

6. Ln. 185: there is a parenthesis that is not open before.

Response: Removed parenthesis ")".

7. Ln. 229: full stop missing after the parenthesis.

Response: Included full stop "." after "(2019)" in "…is based on Deckers et al. (2019). The…".

8. Eq. (1) and (2) vectors should be distinguished from scalar by using bold characters.

Response: The vectors included in Equations 1 and 2 were formatted as bold characters.

9. Ln. 336: which are cases 2 and 3?

Response: The modelling cases 2 and 3 are indicated right after the paragraph; line 341 "Model 2: Thermal conduction", and line 348 "Model 3: Heat-transport without faults". In line 335, "modelling cases" is changed to "model 2 and model 3 (see below)" for clarification.